# LensAge index as a deep learning-based biological age for self-monitoring the risks of age-related diseases and mortality

Ruiyang Li[1,7], Wenben Chen[1,7], Mingyuan Li[1,7], Ruixin Wang[1], Lanqin Zhao[1], Yuanfan Lin [1], Xinwei Chen[1], Yuanjun Shang[1], Xueer Tu[1], Duoru Lin[1], Xiaohang Wu[1], Zhenzhe Lin[1], Andi Xu[1], Xun Wang[1], Dongni Wang[1], Xulin Zhang[1], Meimei Dongye[1], Yunjian Huang[1], Chuan Chen[2], Yi Zhu[3], Chunqiao Liu[1], Youjin Hu [1], Ling Zhao[1], Hong Ouyang [1], Miaoxin Li [4,5], Xuri Li [1] & Haotian Lin [1,4,6] ✉

Age is closely related to human health and disease risks. However, chronologically defined age often disagrees with biological age, primarily due to genetic and environmental variables. Identifying effective indicators for biological age in clinical practice and self-monitoring is important but currently lacking. The human lens accumulates age-related changes that are amenable to rapid and objective assessment. Here, using lens photographs from 20 to 96-year-olds, we develop LensAge to reflect lens aging via deep learning. LensAge is closely correlated with chronological age of relatively healthy individuals ($R^2 > 0.80$, mean absolute errors of 4.25 to 4.82 years). Among the general population, we calculate the LensAge index by contrasting LensAge and chronological age to reflect the aging rate relative to peers. The LensAge index effectively reveals the risks of age-related eye and systemic disease occurrence, as well as all-cause mortality. It outperforms chronological age in reflecting age-related disease risks ($p < 0.001$). More importantly, our models can conveniently work based on smartphone photographs, suggesting suitability for routine self-examination of aging status. Overall, our study demonstrates that the LensAge index may serve as an ideal quantitative indicator for clinically assessing and self-monitoring biological age in humans.

Assessing an individual's aging process is important to evaluate one's health status. As one ages, the human body becomes frail with regard to biological functions and the occurrence of chronic diseases, such as Alzheimer's disease[1], cancer[2], diabetes[3], and cardiovascular diseases[4].

Chronological age is defined as the time that an individual has experienced since birth. Since aging involves complex determinants, including genetic regulation, and the nutritional and environmental factors, peers with the same chronological age vary in aging and may

[1]State Key Laboratory of Ophthalmology, Zhongshan Ophthalmic Center, Sun Yat-sen University, Guangdong Provincial Key Laboratory of Ophthalmology and Vision Science, Guangdong Provincial Clinical Research Center for Ocular Diseases, Guangzhou, China. [2]Sylvester Comprehensive Cancer Center, University of Miami Miller School of Medicine, Miami, FL, USA. [3]Department of Molecular and Cellular Pharmacology, University of Miami Miller School of Medicine, Miami, FL, USA. [4]Center for Precision Medicine and Department of Genetics and Biomedical Informatics, Zhongshan School of Medicine, Sun Yat-sen University, Guangzhou, China. [5]Key Laboratory of Tropical Disease Control (Sun Yat-sen University), Ministry of Education, Sun Yat-sen University, Guangzhou, China. [6]Hainan Eye Hospital and Key Laboratory of Ophthalmology, Zhongshan Ophthalmic Center, Sun Yat-sen University, Haikou, China. [7]These authors contributed equally: Ruiyang Li, Wenben Chen, Mingyuan Li. ✉e-mail: linht5@mail.sysu.edu.cn

have different health status and life expectancy[5–7]. Thus, chronological age does not precisely reveal the true physiological age of individuals.

Biological age assessment based on various physiological biomarkers can quantitatively evaluate the degree of aging and predict the mortality and incidence of age-related diseases more accurately than chronological age. However, measuring biological age is challenging, largely due to obstacles in sample collection, variable aging rates of different tissues, and insufficient reliability of measuring tools and protocols[8]. Intensive investigations of the biological indicators reflecting the overall aging pace of the human body are currently underway. For example, invasive methods measuring telomere length[9] and DNA methylation status[10], profiling transcriptomics[11] and proteomics[12], and the inflammatory aging clock[13] have been used to generate biomarkers of aging at the molecular level using human blood cells. Furthermore, noninvasive techniques using machine learning and medical imaging, such as chest X-ray[14], magnetic resonance imaging (MRI) of the brain[15], and 3D facial imaging[16], were introduced to evaluate biological aging. However, these techniques are limited by high costs or instability in clinical practice. Therefore, a more objective, reliable, convenient, and noninvasive method that can accurately evaluate the biological age of an individual has yet to be developed for broader applications and self-management of health status.

The human lens, located in the anterior segment of the eye, is transparent under normal conditions and exchanges substances with the vitreous through the aqueous humor cycle[17]. Age-dependent changes in the lens include nucleus enlargement, elasticity reduction, and increased opacity, all of which can be objectively and reliably observed through noninvasive imaging and rapidly assessed using digital photography[18]. Thus, the human lens appears to be an optimal tissue with unique advantages for assessing biological age.

In this study, we used informative lens photographs to generate LensAge as an innovative indicator to reveal aging status of lens based on deep learning (DL) models. Under ideal physiological conditions (both genetic and environmental), biological age should be synchronized with chronological age. While in reality, there are almost always differences between biological age and chronological age, which is considered to result from individually different aging processes[19]. Therefore, we measured the difference between LensAge and chronological age as the LensAge index to assess an individual's aging rate relative to peers, and investigated its ability to evaluate the risks of age-related disease occurrence and all-cause mortality. Importantly, we tested whether our models can be generalized to smartphone-based lens photographs, which may have potential applications for self-monitoring the risks of age-related diseases and mortality during aging.

## Results

### Performance of DL models for age estimation

In this study, to generate LensAge, we first developed DL-based age estimation models on a reference dataset of relatively healthy individuals who did not report any medical history of systemic diseases and had no abnormalities in physical examination at baseline. A total of 8255 lens photographs (4542 for diffuse-light mode and 3713 for slit-lamp mode) from 1990 relatively healthy individuals aged between 20 and 96 years (mean age [± s.d.] of 55.3 [± 18.0] years, 63.2% females, Table 1) were included. LensAge at the individual level was calculated by averaging the LensAge values of all diffuse-light or slit-lamp images corresponding to one individual. Four classic convolutional neural networks (CNNs), including InceptionV3, ResNet50, DenseNet, and InceptionResNetV2, were trained to predict the ages of relatively healthy individuals. The most outperformed network was selected for further analyses. The study workflow is summarized in Fig. 1.

Among the four trained CNNs, the InceptionV3 models displayed best overall performance for model validation (Supplementary Table 1). For diffuse-light mode, at the image level, the InceptionV3 model had a mean absolute error (MAE) of 4.88 years (Supplementary

**Table 1 | Baseline characteristics of the datasets**

| | Traditional slit-lamp images | | Smartphone images | |
|---|---|---|---|---|
| | Reference dataset | Analysis dataset | Reference dataset | Analysis dataset |
| **No. of participants** | 1990 | 3433 | 50 | 102 |
| **Nationality, n (%)** | | | | |
| Chinese | 1952 (98.1%) | 3370 (98.2%) | 50 (100%) | 99 (97.1%) |
| Non-Chinese | 38 (1.9%) | 63 (1.8%) | 0 | 3 (2.9%) |
| **Age in years (mean ± s.d.)** | 55.3 ± 18.0 | 66.0 ± 11.5 | 64.6 ± 11.4 | 62.0 ± 10.8 |
| **Distribution of chronological age, n (%)** | | | | |
| ≥ 20 and < 30 | 245 (12.3%) | 8 (0.2%) | 0 | 0 |
| ≥ 30 and < 40 | 231 (11.6%) | 55 (1.6%) | 3 (6.0%) | 4 (3.9%) |
| ≥ 40 and < 50 | 246 (12.4%) | 224 (6.5%) | 0 | 6 (5.9%) |
| ≥ 50 and < 60 | 292 (14.7%) | 624 (18.2%) | 15 (30.0%) | 37 (36.3%) |
| ≥ 60 and < 70 | 507 (25.5%) | 1139 (33.2%) | 12 (24.0%) | 28 (27.5%) |
| ≥ 70 and < 80 | 334 (16.8%) | 976 (28.4%) | 18 (36.0%) | 23 (22.5%) |
| ≥ 80 | 134 (6.7%) | 407 (11.9%) | 2 (4.0%) | 4 (3.9%) |
| **Sex, n (%)** | | | | |
| Male | 732 (36.8%) | 1482 (43.2%) | 20 (40.0%) | 45 (44.1%) |
| Female | 1258 (63.2%) | 1951 (56.8%) | 30 (60.0%) | 57 (55.9%) |
| **Images, n** | | | | |
| Diffuse-light images | 4542 | 5641 | N/A | N/A |
| Slit-lamp images | 3713 | 5663 | 157 | 389 |

Table 1), while at the individual level, the model showed a strong correlation ($R^2 = 0.89$, $p < 1.00e\text{-}36$, Fig. 2a) between LensAge and chronological age, with an MAE of 4.25 years (Supplementary Table 1). For slit-lamp mode, at the image level, the InceptionV3 model had an MAE of 5.25 years (Supplementary Table 1), whereas at the individual level, the model achieved a significant correlation ($R^2 = 0.82$, $p < 1.00e\text{-}36$, Fig. 2c) between LensAge and chronological age, with an MAE of 4.82 years (Supplementary Table 1). In addition, we assessed the level of agreement between InceptionV3-generated LensAge and chronological age using Bland–Altman plots. Bland–Altman plot analyses revealed that the average differences between LensAge and chronological age were −0.21 (−1.96 s.d. −11.40 to +1.96 s.d. 10.98) for diffuse-light mode and −0.59 (−1.96 s.d. −12.79 to +1.96 s.d. 11.62) for slit-lamp mode (Fig. 2b, d). These results demonstrate that the InceptionV3 models show small biases and good performance in age estimation for relatively healthy individuals.

Furthermore, our InceptionV3 models exhibited favorable performance across various types of cataracts (Supplementary Table 2). Although the majority of participants in this study were of Chinese descent, we also included individuals from different nationalities. The performance of the InceptionV3 models in this non-Chinese population demonstrates their potential applicability to other ethnicities (Supplementary Fig. 1). Additionally, we compared the accuracy of our age estimation models with biological age measurements in previous studies and found that the LensAge models achieved reasonably accurate performance among relatively healthy individuals (Supplementary Table 3). Hence, the InceptionV3 models were employed for further analyses.

### Interpretation of the DL age estimation models

We enhanced the interpretability of our DL age estimation models by visualizing the attention regions that the models utilized to extract relevant features from the images. Representative attention maps for

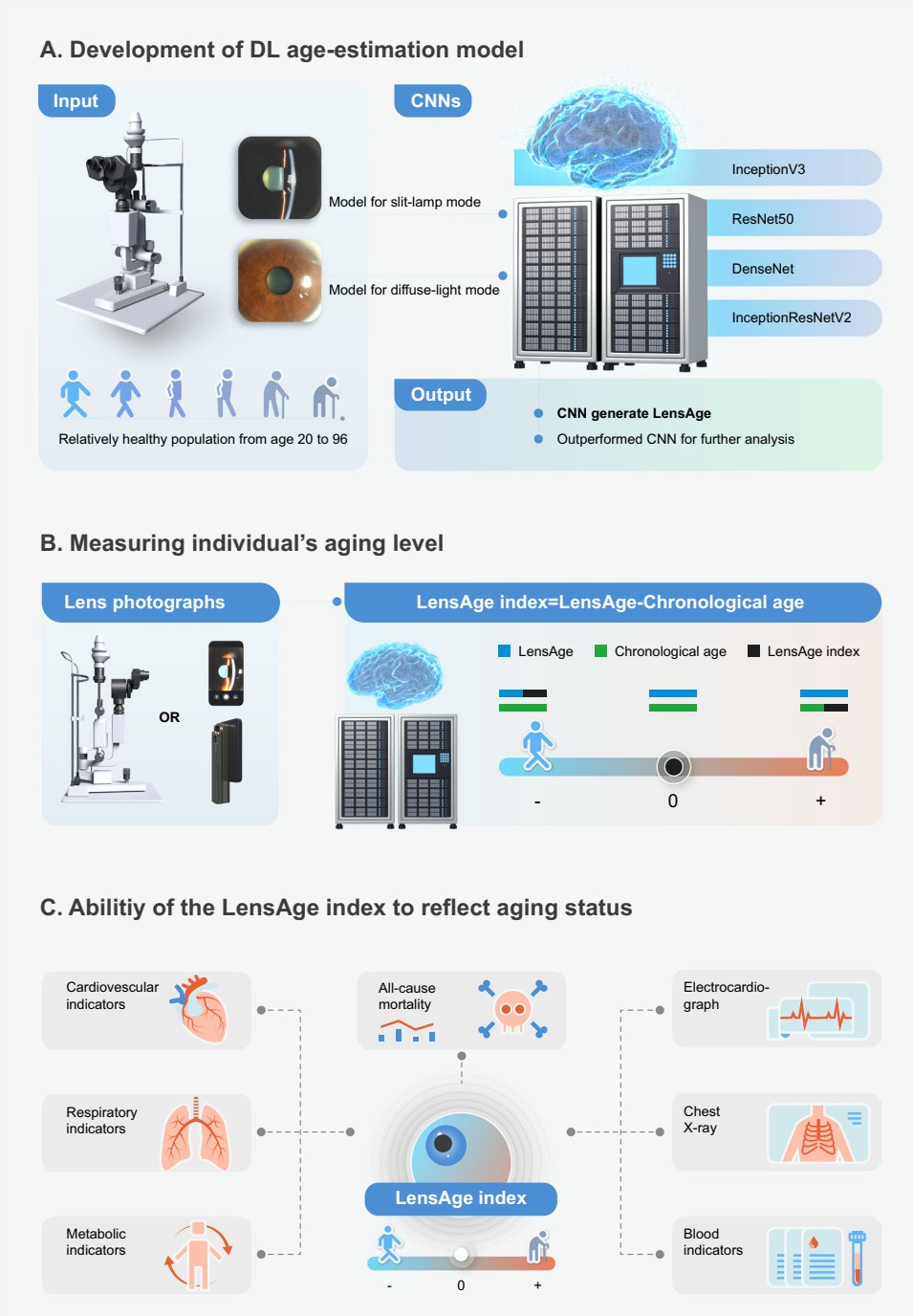

**Fig. 1 | Overview of the study workflow. A** DL-based age estimation models were developed to predict LensAge using lens photographs taken in diffuse-light or slit-lamp mode from relatively healthy individuals aged 20 to 96 years with chronological-age labels. **B** DL-based age estimation models can be applied to lens photographs taken with traditional slit lamps and smartphones. The difference between LensAge at the individual level and chronological age was considered a measure of an individual's aging level and served as the LensAge index. A positive LensAge index indicates accelerated aging relative to peers of the same chronological age, while a negative LensAge index indicates decelerated aging or a younger biological age compared to peers. **C** The ability of the LensAge index to assess risks of ocular and systemic aging conditions and all-cause mortality was analyzed. DL, deep learning; CNN, convolutional neural network.

age estimation are shown in Fig. 2e, f. The specific lens areas highlighted by the heatmaps indicate that the DL models prioritize the lens during age assessment for both diffuse-light and slit-lamp modes. These results suggest that our DL age estimation models can extract information on the aging characteristics of human lenses across different age groups.

Furthermore, we analyzed the influence of masking different lens structures on LensAge prediction for slit-lamp mode. When masking the lens cortex for cortical and noncortical cataracts, the predictive error was higher among cortical cataracts (adjusted odds ratio [OR] = 1.23, 95% confidence interval [CI] 1.16–1.31, $p = 6.72e-11$, Supplementary Table 4). This indicates that in the case of cortical cataracts, the lens cortex regions played a more crucial role in the decision-making process of the model compared to other types of cataracts. Similarly, the predictive errors were higher among nuclear cataracts and subcapsular cataracts than among other types of cataracts when masking the lens

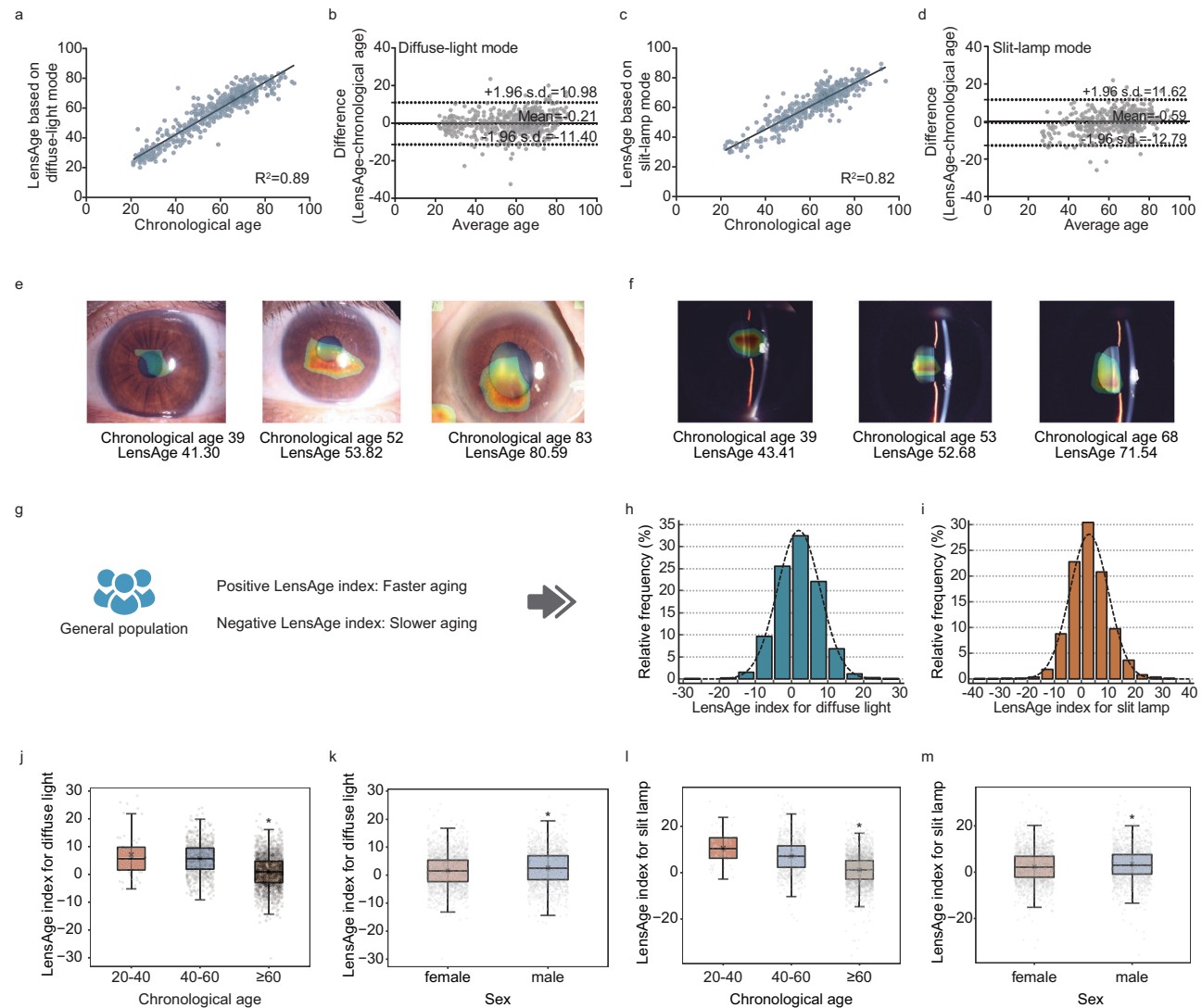

**Fig. 2 | Analyses of LensAge and the LensAge index based on InceptionV3 models. a**, **c** Scatterplots for the correlation of LensAge at the individual level with chronological age of relatively healthy participants for diffuse-light mode (**a**) and slit-lamp mode (**c**) ($p < 1.00e\text{-}36$, two-sided linear regressions; $n = 525$ for diffuse-light mode; $n = 430$ for slit-lamp mode). **b**, **d** Bland–Altman plots for the agreement between LensAge at the individual level and chronological age of relatively healthy participants for diffuse-light mode (**b**) and slit-lamp mode (**d**) ($n = 525$ for diffuse-light mode; $n = 430$ for slit-lamp mode). **e**, **f** Attention maps for age estimation of the model based on lens photographs from different individuals for diffuse-light mode (**e**) and slit-lamp mode (**f**). **g** The LensAge index was used to measure an individual's aging level among the general population. **h**, **i** The distributions of the LensAge index for diffuse-light mode (**h**) and slit-lamp mode (**i**) among the general population. **j**, **l** The distributions of the LensAge index for diffuse-light mode (**j**) and slit-lamp mode (**l**) by age groups. The mean LensAge indexes in the groups aged 60 years and above were less than those for the other age groups (*$p < 0.001$; diffuse-

light mode: 20–40 vs. ≥ 60, $p = 3.20e\text{-}15$, mean difference = 6.10, 95% CI 4.59–7.61, $n = 2464$; 40–60 *vs.* ≥ 60, $p < 1.00e\text{-}36$, mean difference = 4.76, 95% CI 4.32–5.20, $n = 3,205$; slit-lamp mode: 20–40 *vs.* ≥ 60, $p = 2.84e\text{-}29$, mean difference = 9.34, 95% CI 7.73–10.95, $n = 2483$; 40–60 *vs.* ≥ 60, $p < 1.00e\text{-}36$, mean difference = 5.95, 95% CI 5.43–6.46, $n = 3237$; two-sided Student's t tests). **k**, **m** The distributions of the LensAge index for diffuse-light mode (**k**) and slit-lamp mode (**m**) by sex groups. The mean LensAge indexes for the males were greater than those for the females (*$p < 0.001$; diffuse-light mode, $p = 3.00e\text{-}6$, mean difference = 0.97, 95% CI 0.56–1.38, $n = 3,258$; slit-lamp mode, $p = 2.40e\text{-}5$, mean difference = 1.05, 95% CI 0.56–1.54, $n = 3298$; two-sided Student's t-tests). The cross symbols denote the means, the thick central lines and triangle symbols denote the medians, the lower and upper box limits denote the lowest and third quartiles, and the whiskers extend from the box to the outermost extreme values but no more than 1.5 times the interquartile range in panel (**j**–**m**). Source data are provided as a Source Data file.

nucleus and lens capsule, respectively (lens nucleus, adjusted OR = 1.24, 95% CI 1.16–1.33, $p = 1.55e\text{-}10$; lens capsule, adjusted OR = 1.12, 95% CI 1.02–1.22, $p = 1.60e\text{-}2$; Supplementary Table 4). These results indicate that our models can focus on specific lens regions corresponding to different types of cataracts to make accurate age estimations.

**Measurement of aging progression in the general population**
We subsequently expanded our analyses to encompass a general population consisting of both healthy and unhealthy participants and assessed LensAge within the analysis dataset, including 3433 participants (mean age [± s.d.] of 66.0 [± 11.5] years, 56.8% females)

aged 20 to 96 years (Table 1). The individual-level discrepancy between LensAge and chronological age indicated that the aging process deviated from the aging norm learned by DL models for peers of the same chronological age in the reference dataset. This difference, served as the LensAge index, provided a measure of an individual's aging progression and highlighted its deviation from the norm. A positive LensAge index indicated that an individual was older than peers, whereas a negative index indicated younger (Figs. 1b, 2g).

The distributions of the LensAge index in the analysis dataset are shown in Fig. 2h–i and Supplementary Table 5. For the population

photographed by diffuse light, the mean (± s.d.) and median (inter-quartile range [IQR]) of the LensAge index were 2.0 (± 5.9) and 1.9 (−2.0, 6.0), respectively. The proportions of fast agers with a LensAge index greater than 5, 10, and 20 years were 30.5%, 8.3%, and 0.4%, respectively. For the slit-lamp images, a similar trend was observed. The mean (± s.d.) and median (IQR) of the LensAge index were 2.9 (± 7.1) and 2.5 (−1.7, 7.2), respectively. The proportions of fast agers with a LensAge index greater than 5, 10, and 20 years were 35.5%, 14.7%, and 1.3%, respectively.

We evaluated the LensAge index by stratifying the age groups into 20–40, 40–60, and ≥60 years. For both diffuse-light and slit-lamp modes, the mean LensAge indexes for the groups older than 60 years were less than those for the other age groups (diffuse-light mode: 20–40 vs. ≥60, $p = 3.20e-15$, mean difference = 6.10, 95% CI 4.59–7.61; 40–60 vs. ≥60, $p < 1.00e-36$, mean difference = 4.76, 95% CI 4.32–5.20, Fig. 2j; slit-lamp mode: 20–40 vs. ≥60, $p = 2.84e-29$, mean difference = 9.34, 95% CI 7.73–10.95; 40–60 vs. ≥60, $p < 1.00e-36$, mean difference = 5.95, 95% CI 5.43–6.46, Fig. 2l), which demonstrates that the aging diversity may reduce at older ages. When stratifying by sex, the mean LensAge indexes for males were greater than those for females (diffuse-light mode, $p = 3.00e-6$, mean difference = 0.97, 95% CI 0.56–1.38, Fig. 2k; slit-lamp mode, $p = 2.40e-5$, mean difference = 1.05, 95% CI 0.56–1.54, Fig. 2m).

### The ability of the LensAge index to reflect the risks of ocular age-related diseases

In order to investigate whether the LensAge index can be an effective marker of biological age, we assessed its ability to reveal the risks of ocular age-related conditions in the analysis dataset using logistic models adjusted for demographic (chronological age, sex, race, region, and occupation) and lifestyle (smoking and alcohol intake status) covariates. The individuals were stratified into two groups for further analyses: those under 60 years old and those aged 60 years and above.

Among individuals of all age groups, those with a positive LensAge index had a higher risk of moderate or severe visual impairment (adjusted OR = 1.65, 95% CI 1.31–2.08, $p = 1.80e-5$), senile cataracts (adjusted OR = 1.76, 95% CI 1.45–2.14, $p = 1.50e-8$), and vitreous opacity (adjusted OR = 1.89, 95% CI 1.27–2.81, $p = 1.84e-3$), compared to those with a negative LensAge index for diffuse-light mode (Fig. 3a). Among individuals of all age groups with a positive LensAge index, an increase in the LensAge index was positively associated with the occurrence of moderate or severe visual impairment (adjusted OR = 1.14, 95% CI 1.10-1.19, $p = 9.25e-12$), senile cataracts (adjusted OR = 1.14, 95% CI 1.10–1.19, $p = 5.29e-13$), and vitreous opacity (adjusted OR = 1.06, 95% CI 1.03-1.10, $p = 4.06e-4$) for diffuse-light mode (Fig. 3b). Most of the findings were also consistent for slit-lamp mode (Fig. 3a, b) and across different age groups (Supplementary Tables 6–9). Compared to individuals with LensAge results of < 60 years, those with LensAge results of ≥ 60 years had a higher risk of moderate or severe visual impairment (slit-lamp mode, adjusted OR = 1.72, 95% CI 1.24–2.38, $p = 1.20e-3$) and senile cataracts (diffuse-light mode, adjusted OR = 2.26, 95% CI 1.71–2.99, $p = 1.03e-8$; slit-lamp mode, adjusted OR = 1.97, 95% CI 1.49–2.60, $p = 1.95e-6$) (Supplementary Table 10). These findings indicate that the LensAge index can be an effective measure of biological age to reflect risks of ocular age-related diseases and advanced aging in the human eyes.

### The ability of the LensAge index to reflect the risks of systemic age-related diseases

We next assessed the ability of the LensAge index to evaluate the risks of systemic age-related diseases using adjusted logistic models and blood glucose (BG) levels using adjusted linear regression models in the analysis dataset. Subgroup analyses were also performed by dividing participants into two groups: those under 60 years old and those aged 60 years and above.

Among individuals of all age groups, those with a positive LensAge index had a higher risk of systemic age-related diseases (diabetes, hypertension, coronary heart disease, cancer and cerebral infarction) (adjusted OR = 1.26, 95% CI 1.05–1.52, $p = 0.010$), age-related changes in chest X-ray findings (arteriosclerosis and left ventricular hypertrophy) (adjusted OR = 1.50, 95% CI 1.17–1.92, $p = 0.010$), and age-related changes in electrocardiographic findings (myocardial ischemia, myocardial infarction, atrial fibrillation, and hypertensive heart disease) (adjusted OR = 1.27, 95% CI 1.08–1.50, $p = 0.004$), compared to those with a negative LensAge index for diffuse-light mode (Fig. 3a). In addition, to further evaluate the impact of significantly higher LensAge index on biological aging, we compared the differences in the risks of age-related diseases between individuals with a LensAge index in the highest quartile and at the moderate level (in the second or third quartiles). The results show that individuals with the LensAge index in the highest quartile had a higher risk of age-related diseases than those with the LensAge index at the moderate level (diffuse-light mode, adjusted OR = 1.61, 95% CI 1.31–1.97, $p = 5.46e-6$; slit-lamp mode, adjusted OR = 1.23, 95% CI 1.00–1.53, $p = 4.73e-2$; Supplementary Table 11). Among the individuals of all age groups with a positive LensAge index for diffuse-light mode, the increase in the LensAge index was positively correlated with the occurrence of age-related diseases (adjusted OR = 1.10, 95% CI 1.07–1.13, $p = 5.97e-11$), age-related chest X-ray findings (adjusted OR = 1.06, 95% CI 1.02–1.10, $p = 0.001$), and electrocardiographic findings (adjusted OR = 1.03, 95% CI 1.00–1.05, $p = 0.042$) (Fig. 3b). The above findings were also mostly consistent for slit-lamp mode (Fig. 3a, b) and across different age groups (Supplementary Tables 6–9). Moreover, the degree of positivity of the LensAge index was significantly correlated with the increase in BG (diffuse-light mode, adjusted $\beta = 0.04$, 95% CI 0.01–0.07, $p = 0.006$, Supplementary Table 12). Additionally, compared with individuals with LensAge results of < 60 years old, those with LensAge results of ≥ 60 years old had a higher risk of age-related diseases (diffuse-light mode, adjusted OR = 2.00, 95% CI 1.48–2.69, $p = 5.09e-6$; slit-lamp mode, adjusted OR = 1.36, 95% CI 1.02–1.79, $p = 0.034$; Supplementary Table 10).

We further compared the ability of the LensAge index to predict the occurrence of age-related diseases with that of chronological age using receiver operating characteristic (ROC) curve analyses, particular among individuals with a significantly greater LensAge index. We performed the ROC analyses among the participants with a LensAge index in the lowest quartile or the highest quartile. The ROC curves graphically illustrate the predictive performance of the LensAge index with an area under the curve (AUC) of 0.621 (95% CI 0.596–0.645) for diffuse-light mode (Fig. 3c), and 0.600 (95% CI 0.575–0.624) for slit-lamp mode (Fig. 3d). Compared with those of chronological age, the AUCs of the LensAge index were significantly greater (diffuse-light mode, difference in AUCs = 0.098, 95% CI 0.077–0.120, $Z = 8.968$, $p < 0.0001$, Fig. 3c; slit-lamp mode, difference in AUCs = 0.090, 95% CI 0.030–0.151, $Z = 2.924$, $p = 3.50e-3$, Fig. 3d), demonstrating that the LensAge index can better reflect the aging process in humans and can be an optimized indicator of age-related disease risks in whole body.

### The predictive performance of the LensAge index for all-cause mortality

To comprehensively assess the LensAge index's ability to indicate biological age, we evaluated the predictive performance of the LensAge index for all-cause mortality using Cox proportional hazards regression models adjusted for chronological age, sex, race, region, occupation, smoking, and alcohol intake status. The results show that each 1-year increase in the LensAge index for the diffuse-light mode was associated with an 8% relative increase in the risk of all-cause mortality (adjusted hazard rate [HR] = 1.08, 95% CI 1.01–1.15, $p = 2.43e-2$; Supplementary Table 13). Comparatively, individuals in the second quartile of the LensAge index had a similar mortality risk to those in the

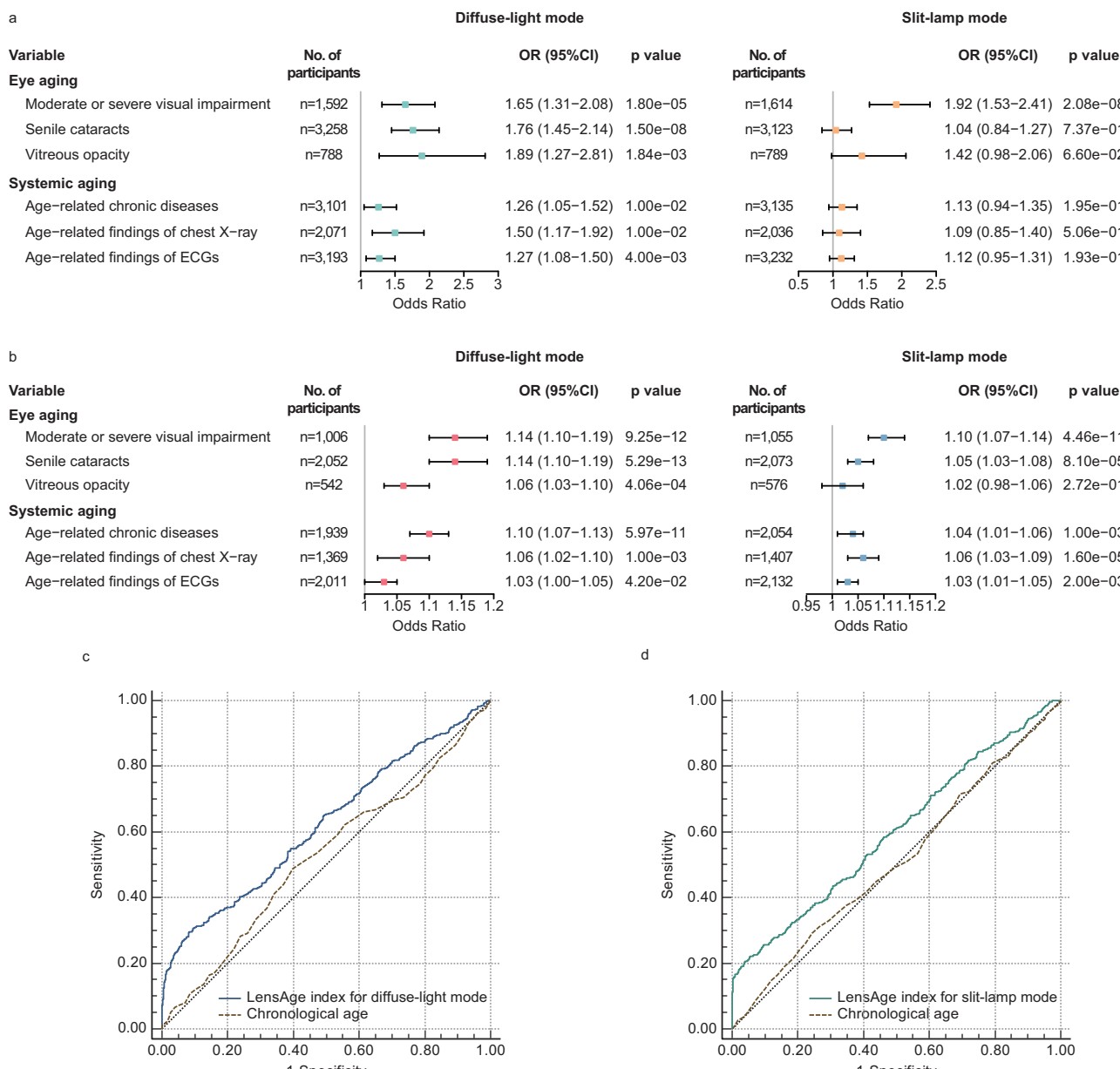

**Fig. 3 | The ability of the LensAge index to evaluate age-related disease risks.**
**a** Comparison of age-related changes between individuals with positive and negative LensAge indexes. *p* values from two-sided tests using adjusted logistic regressions. **b** Association of the LensAge index with age-related changes in individuals with a positive LensAge index. *p*-values from two-sided tests using adjusted logistic regressions. **c** Comparison of the AUCs between the LensAge index and chronological age in predicting the occurrence of age-related diseases for diffuse-light mode among the participants with a LensAge index in the lowest quartile or the highest quartile using ROC curves. LensAge index, AUC = 0.621 (95% CI 0.596–0.645); chronological age, AUC = 0.523 (95% CI 0.498–0.548); difference in

AUCs = 0.098 (95% CI 0.077–0.120), *Z* = 8.968, *p* < 0.0001, *n* = 1555, two-sided paired DeLong test. **d** Comparison of the AUCs between the LensAge index and chronological age in predicting the occurrence of age-related diseases for slit-lamp mode among the participants with a LensAge index in the lowest quartile or the highest quartile using ROC curves. LensAge index, AUC = 0.600 (95% CI 0.575–0.624); chronological age, AUC = 0.509 (95% CI 0.484–0.535); difference in AUCs = 0.090 (95% CI 0.030–0.151), *Z* = 2.924, *p* = 3.50e-3, *n* = 1536, two-sided paired DeLong test. Error bars show 95% CIs for OR values in (**a**, **b**). OR Odds ratio, CI Confidence interval, ROC Receiver operating characteristic, AUC Area under the curve. Source data are provided as a Source Data file.

lowest quartile (adjusted HR = 1.11, 95% CI 0.47–2.63, *p* = 0.818; Fig. 4a). Notably, participants in the third and fourth quartiles of the LensAge index had significantly increased all-cause mortality risks compared to those in the lowest quartile (adjusted HR = 2.55, 95% CI 1.22–5.37, *p* = 0.013; adjusted HR = 2.96, 95% CI 1.39–6.32, *p* = 0.005; respectively; Fig. 4a). Similar results were observed for the survival analyses conducted using the slit-lamp mode (Fig. 4b and Supplementary Table 13). These findings demonstrate that the LensAge index is a significant predictor of survival, and the acceleration or deceleration of aging

detected by LensAge measurements aligns with individual aging status.

## Assessment of lens aging using smartphones
We further implemented LensAge to evaluate biological aging using smartphone photographs (Fig. 5a). A total of 389 smartphone images from 102 participants (mean age [± s.d.] of 62.0 [± 10.8] years, 55.9% females, Table 1) were included in the analysis dataset of smartphone images. Among them, 157 images from 50 participants (mean age

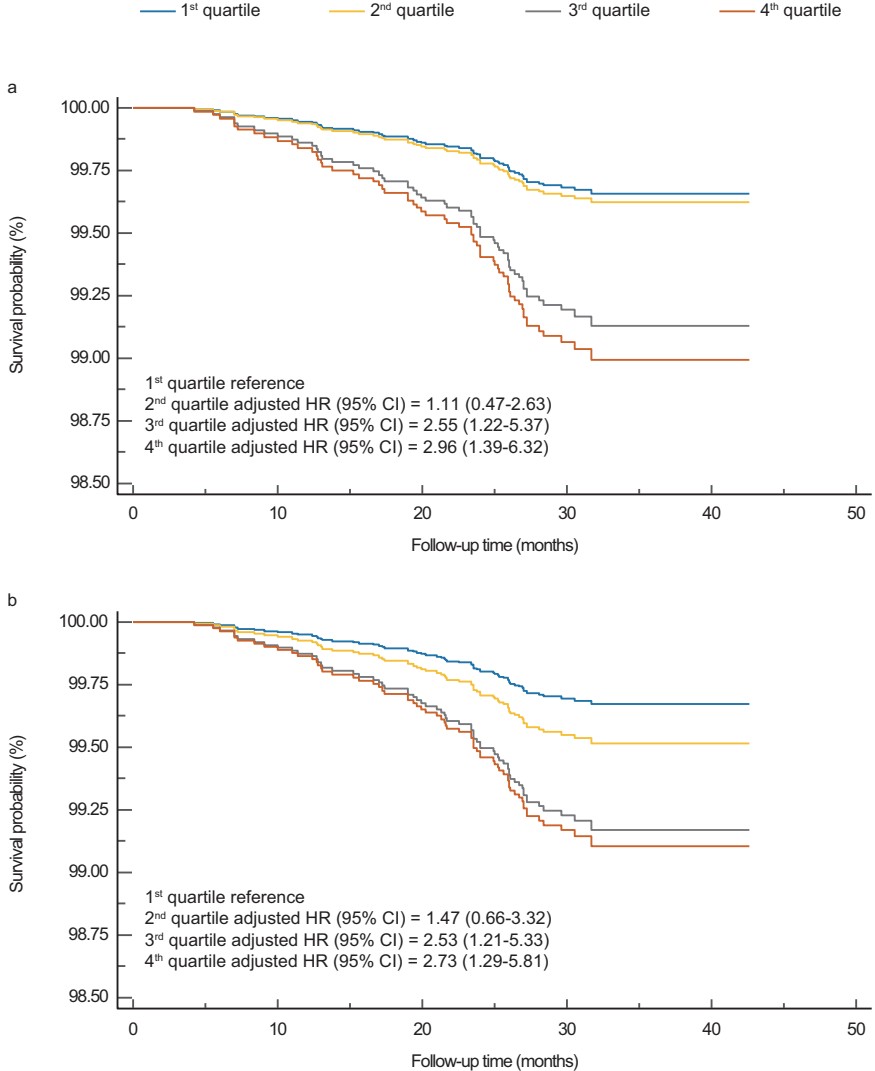

**Fig. 4 | Survival curves for all-cause mortality risk by LensAge index quartiles.** Mortality risks are shown over time for participants in different LensAge index quartiles for diffuse-light mode (**a**) and slit-lamp mode (**b**) (*n* = 2834 for diffuse-light mode; *n* = 2868 for slit-lamp mode). Comparatively, individuals in the second quartile of the LensAge index had a similar mortality risk to those in the lowest quartile (*p* = 0.818 for diffuse-light mode; *p* = 0.348 for diffuse-light mode).

Individuals in the third and fourth quartiles of the LensAge index had significantly relative increased all-cause mortality risks compared to those in the lowest quartile (diffuse-light mode: 3rd quartile *p* = 0.013, 4th quartile *p* = 0.005; slit-lamp mode: 3rd quartile *p* = 0.014, 4th quartile *p* = 0.009). *p*-values from two-sided tests using adjusted Cox proportional hazards regressions. HR Hazard ratio, CI Confidence interval. Source data are provided as a Source Data file.

[± s.d.] of 64.6 [± 11.4] years, 60.0% females) without a medical history of systemic diseases were used as a reference dataset of relatively healthy individuals for model accuracy estimation.

LensAge generated by the DL model based on smartphone photographs had a strong correlation ($R^2$ = 0.71 at the individual level, *p* = 1.59e-14, Fig. 5b, c) with chronological age in the reference dataset, with MAEs of 6.87 years at the image level and 6.80 years at the individual level. In the analysis dataset, compared to those with a negative LensAge index, those with a positive LensAge index had a higher risk of age-related chronic diseases (diabetes, hypertension, coronary heart disease, and cancer) (adjusted OR = 4.21, 95% CI 1.44–12.36, *p* = 0.009, Fig. 5d). Among these individuals with a positive LensAge index, the LensAge index was positively associated with the occurrence of age-related diseases (adjusted OR = 1.53, 95% CI 1.09–2.15, *p* = 0.013, Fig. 5d). Thus, the LensAge index based on smartphone photographs can be an effective indicator of biological age for efficient self-examination of disease risks and health status during aging.

## Discussion

In this study, we developed DL models that evaluated LensAge using lens photographs of a relatively healthy population aged 20 to 96 years. Based on LensAge, we calculated the difference between LensAge and chronological age as the LensAge index to reflect individual's aging level relative to peers. The LensAge index can be applied to reveal the risks of age-related diseases, including eye aging conditions and systemic age-related diseases, and to predict all-cause mortality. Furthermore, compared to chronological age, the LensAge index achieved superior performance of assessing risks of age-related diseases in humans. Importantly, our models can also be conveniently implemented based on smartphone photographs for biological age assessment, showing great potential to be used in self-monitoring aging status. Our results show that the LensAge index is able to signify age-related changes and can be an optimized indicator of biological age.

Our interpretation analyses revealed the significance of opacities in different lens structures for the decision-making process of our DL

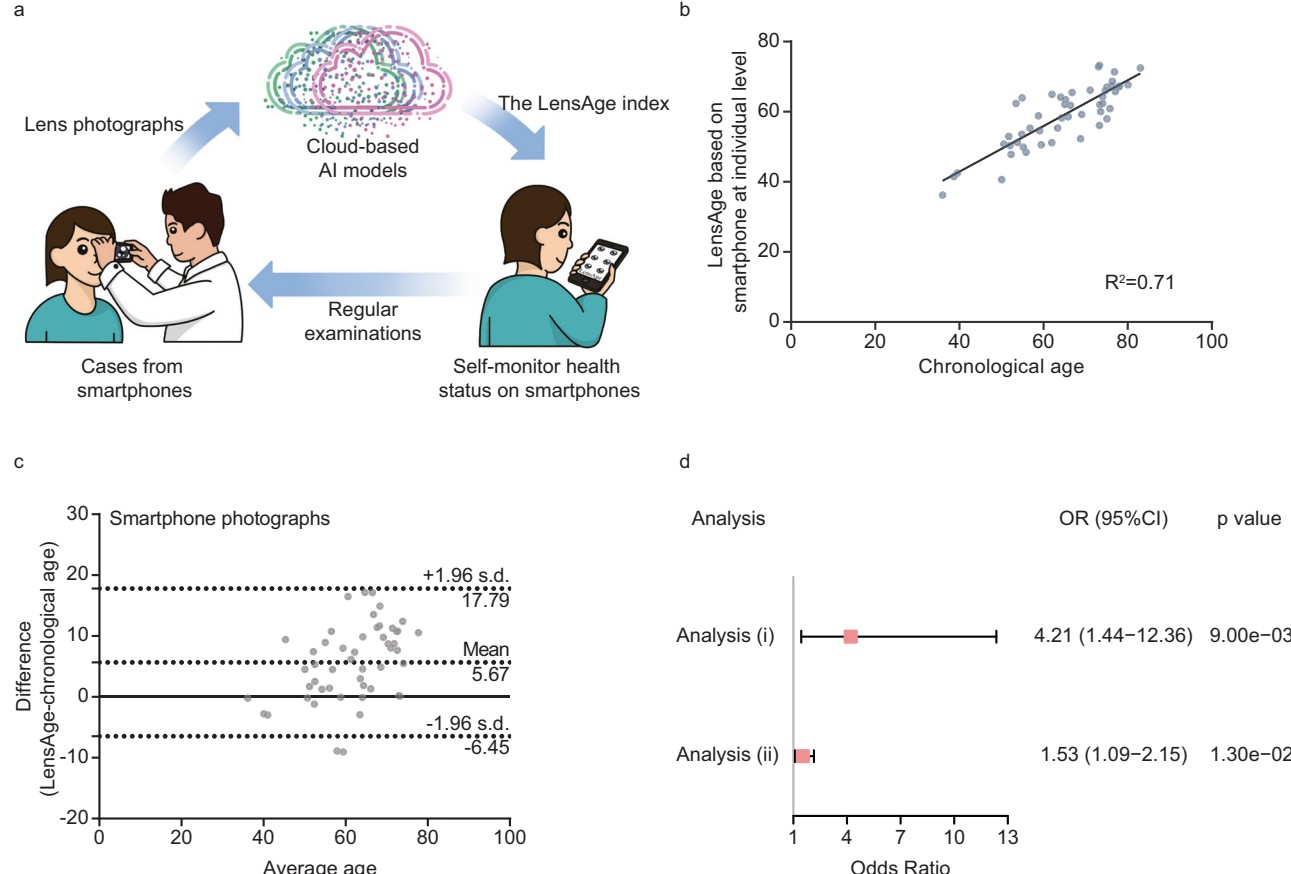

**Fig. 5 | Assessment of lens aging using smartphone photographs. a** The application of the DL-based age estimation model on smartphones for self-monitoring aging status. **b** Scatterplot shows correlation of LensAge at the individual level determined using smartphone photographs with chronological age among relatively healthy participants ($p = 1.59e-14$, two-sided linear regression, $n = 50$). **c** Bland–Altman plot shows agreement between LensAge at the individual level using smartphone photographs and chronological age among relatively healthy individuals. ($n = 50$). **d** The analyses of the LensAge index to reflect age-related diseases based on smartphone photographs among the general population. Analysis (i): comparison of the occurrence of age-related diseases between individuals with positive and negative LensAge index ($n = 102$). Analysis (ii): association of the LensAge index with the risk of occurrence of age-related diseases among individuals with positive LensAge index ($n = 37$). $p$-values from two-sided tests using adjusted logistic regressions. Error bars show 95% CIs for OR values. AI Artificial intelligence, OR Odds ratio, CI Confidence interval. Source data are provided as a Source Data file.

models. An increase in the LensAge index was positively associated with the occurrence of senile cataracts, demonstrating that the increased opacities in lens play a crucial role in evaluating biological age using our DL models. Moreover, our masking trial demonstrated that our models effectively directed their attention towards the specific regions associated with the corresponding cataract types, showing their objectiveness. However, it should be noted that the LensAge index for slit-lamp mode showed significant correlations only among analyses on certain eye aging diseases and systemic conditions, indicating the weaker ability of slit-lamp mode to reveal aging status compared with that of diffuse-light mode. This may be attributed to inconsistencies in the angles or widths of the slit light, which are crucial for capturing intricate lens features and structures during aging. Further studies should implement additional strategies to capture lens photographs of higher quality, such as introducing a quality control pipeline, to effectively assess and filter out any subpar images.

Compared to invasive methods or other medical imaging techniques used for biological age measurements, our age estimation models achieved favorably accurate performance. Traditional approaches such as analyses of transcriptome[11], DNA methylation[20,21], and blood profiles[22,23] require collecting blood sample or biopsied tissue, making them invasive and limiting their application in large-scale evaluations or routine physical examinations. Furthermore, the use of different platforms to measure omics data may introduce

technical noise to the results[19]. Additionally, medical imaging methods were employed to capture age-related features for age prediction[24,25]. However, the high cost and inconvenient nature of neuroimaging in routine examinations[26], as well as the privacy concerns related to facial imaging[27] have hindered their broader application. Some studies have explored biological age estimation using ocular retinal photographs[28–30]. While a retinal-photograph-based model showed favorable performance[30], its application is limited by the presence of opaque dioptric media that can obscure the fundus, particularly in conditions such as cataracts and vitreous opacity which are common among older populations. Additionally, capturing retinal images still requires professional equipment and expertise from the personnel involved, which poses challenges to their widespread and convenient applications.

Biological age estimation based on lens photographs offers advantages over the previously mentioned methods and holds substantial potential for self-monitoring and large-scale evaluations. Human lens accumulates biological changes during aging that can be objectively observed and are amenable to rapid assessment. In this study, our DL age estimation models successfully learned the population-level characteristics of lens aging and effectively revealed the risks of age-related disease occurrence in eyes. Moreover, as an important structure, human lens is linked to the metabolic cycle of the entire body[31], and it serves as a crucial connection between ocular and

systemic aging. A previous study has demonstrated that the lens is essential for the structural and functional changes of the neurological system during aging[32]. Similarly, the LensAge index generated by our DL models can reflect the risk of systemic diseases and even all-cause mortality. These findings suggest that the human lens could be an excellent tissue for monitoring the aging process in both the eye and whole body. More importantly, lens features can be conveniently and objectively captured by portable devices to identify disease states[33]. This is particularly significant as previous biological age assessments often relied on technologies limited to hospital or laboratory settings and were impractical for community or home use. By utilizing our smartphone-based method, we demonstrated the great potential of assessing biological age for self-monitoring aging status. The rapid development of mobile health (mHealth) and the widespread use of mobile devices allow for continuous monitoring and intervention in medical conditions, anytime and anywhere[34]. Our methods based on smartphone photographs can facilitate regular self-assessment of disease risks and aging/health status. Although our current approach requires attaching a portable slit lamp for capturing lens photographs, it has proven effective in the context of mHealth for self-monitoring aging status. Future studies should validate the utility of smartphone-based LensAge index determination through large-scale prospective trials, while also exploring more convenient and cost-effective alternatives to portable slit lamps.

Currently, most published researches on biological age assessment focus on specific age groups, such as middle-aged or elderly individuals[6,16,29]. Given that age-related changes occur throughout the entire lifespan and manifests at different rates across different age groups[12], we investigated a more diverse population spanning ages 20 to 96 years to provide a more comprehensive understanding. Our findings specifically highlight the significant disparity of the LensAge index among individuals younger than 60 years old. This observation aligns with previous studies investigating the aging process, which have consistently reported smaller variations and more stable states of specific biological age makers among individuals older than 60 years[16,30]. Consequently, it suggests that interventions targeting aging should primarily be initiated during the stage characterized by substantial heterogeneity in the context of aging. Nevertheless, it is important to note that the clinical merit of our models is not diminished within the older age groups. The LensAge index provides a measure of an individual's aging level relative to their peers of the same chronological age. In the population aged more than 60 years, a positive LensAge index showed a significant correlation with an increased risk of age-related diseases (Supplementary Table 6), demonstrating the effectiveness of our models in accurately identifying disease risks within this age group. Therefore, our method provides an effective indicator for revealing the biological age of populations spanning different age groups.

Importantly, the process of aging is multifaceted, involving various aspects such as the decline of bodily functions, the occurrence and progression of age-related diseases, and mortality. In our study, we extended the application of our models to evaluate age-related status and the risk of all-cause mortality in the general population. By adjusting for chronological age as a covariate, we found that the higher LensAge index was significantly correlated with an increased risk of age-related diseases in human eyes. Additionally, we also provided evidence for potential associations between lens aging and systemic aging. Notably, the LensAge index showed a significant correlation with age-related chronic diseases and BG levels, shedding light on the metabolic conditions of individuals. Furthermore, previous studies have applied markers of biological age to predict mortality or estimate time until death, highlighting the potential utility of such markers[14,29,30]. Similarly, through a comprehensive assessment of the predictive performance of the LensAge index for mortality risk, our results indicate that the LensAge index serves as a noteworthy predictor of death in humans, elucidating a portion of the mortality variation that remains unexplained by chronological age itself. This highlights its significant clinical value as an indicator of biological age identifying individuals at high risk of mortality, enabling early interventions. These findings suggest a considerable synchronization between the aging processes in the lens and systemic metabolism, extending to the mortality risk. Therefore, the application of our lens aging assessment technique holds promise for self-monitoring health status and implementing targeted interventions aimed at addressing aging-related concerns.

Our study has a few limitations that should be acknowledged. First, we may have not included individuals with extremely poor health status across various age groups, as they were less likely to participate in our study. Second, although a small proportion of non-Chinese people were included in our study, showing the potential generalizability of our models across additional ethnicities and nationalities, further larger-scale validation on other ethnic groups and countries is needed as an important continuation of this work. Third, our methods may not be appropriate for a proportion of participants with complicated cataracts or medical history of intraocular surgeries. However, it does not impact our primary objective of evaluating the feasibility of utilizing age-related features in lens to reveal biological age for large-scale application and self-monitoring. Fourth, although we may include participants who were unaware of their underlying and undiagnosed diseases in the relatively healthy datasets, we excluded most unhealthy patients based on the medical history and physical examinations, and a sufficient sample size of relatively healthy participants was included to minimize the influence of any patients with underlying conditions on our models to learn the average aging characteristics among the population.

In conclusion, we introduce an innovative DL-based technique, that effectively assesses the risks of age-related diseases and mortality. Importantly, this method can be implemented using smartphone photographs, providing accurate health monitoring capabilities. These findings emphasize the effectiveness of the LensAge index as an indicator of biological age in humans, highlighting its potential for widespread self-monitoring health conditions during the aging process.

## Methods
### Study design and population
For diffuse-light and slit-lamp photographs, participants aged 20 to 96 years were recruited from (1) an ongoing national Chinese cataract screening program by the Chinese Medical Alliance for Artificial Intelligence (CMAAI) between April 2018 and May 2021 with comprehensive baseline information (chronological age, sex, race, region, and occupation), anthropometric and lifestyle factors (not/formerly/currently smoking and alcohol intake status), medical history of diseases, regular physical examinations and ophthalmic examinations; and (2) the retrospective hospital dataset of Zhongshan Ophthalmic Center (Sun Yat-sen University, Guangdong, China) between January 2020 and May 2021 with comprehensive baseline information (chronological age, sex, race, region, and occupation), medical history of diseases, regular physical examinations, ophthalmic examinations, chest X-ray examinations, electrocardiographs, full blood count, and basic profile of blood collected from the hospital admission records. All enrolled participants were eligible for the study if they had no history of previous eye surgery, eye trauma, ocular diseases (high myopia, etc.) that can cause complicated cataracts, and long-term use of corticosteroids or other drugs that can cause drug-induced cataracts. The collected systemic medical histories at baseline included diabetes, hypertension, cardiovascular diseases, cerebrovascular diseases, cancer, and other chronic systemic diseases. All participants underwent regular physical examinations including heart rate, blood pressure, respiratory rate, height, and weight and ophthalmic examinations consisted of functional and structural examinations, including visual acuity, intraocular

pressure, slit-lamp examinations, fundoscopy examinations, and cycloplegic refraction. All participants underwent binocular anterior segment photographs for diffuse-light and slit-lamp modes using a variety of slit lamps, including the BQ-900, BX-900, OVS-II, and PSL-Classic. The distribution of images captured by the different traditional slit lamps is summarized in Supplementary Table 14. For smartphone photographs, participants aged 35 to 90 years were recruited from Sun Yat-sen Memorial Hospital (Sun Yat-sen University, Guangdong, China) and community screening in Lianzhou, Guangdong, China. The inclusion criteria were the same as described above. Baseline information and a comprehensive medical history of diseases were also collected from questionnaires for the recruited individuals. All participants had smartphone photographs taken for both eyes with a portable slit lamp (MediWorks portable slit lamp S150, Shanghai) attached to the iPhone/Huawei smartphones. All enrolled individuals had no more than three photographs taken for each eye for each mode. The images for which the lens area was included with sufficient image quality were eligible for this study. The demographics and summary of the study datasets are shown in Table 1. The overall study workflow is summarized in Fig. 1.

The study protocol was approved by the Institutional Review Board/Ethics Committee of Zhongshan Ophthalmic Center and registered on ClinicalTrials.gov (Identifier NCT05588921). Before data collection, informed consent was obtained from each participant. The investigators followed the requirements of the Declaration of Helsinki throughout the study. All datasets used in the study were deidentified before being transferred to the study investigators.

### DL model development for age estimation

Relatively healthy participants were defined as those who did not report any medical history of systemic diseases and had no physical examination abnormalities at baseline. Lens photographs from relatively healthy participants were included in the reference dataset. A total of 8255 anterior segment photographs, including diffuse-light or slit-lamp mode images of sufficient image quality together with chronological age labels from 1990 relatively healthy individuals (mean age [± s.d.] of 55.3 [ ± 18.0] years, 63.2% female) were included in the training and evaluation processes of the DL age estimation models. The chronological age label was the age of an individual since birth, rounded to the nearest year at the time when the lens photographs were taken. Birth dates were obtained from Chinese government-issued official resident identity cards. Images for model development were split into a training set (60%), a tuning set (20%), and a validation set (20%). All data were split at the individual level, and images belonging to one individual did not appear in different sets, which can avoid overestimation in the evaluation of DL models.

We trained the DL age estimation models based on CNNs using lens photographs of diffuse-light or slit-lamp mode to generate LensAge (Fig. 1a), exploring whether the changes in lens features reflect the process of aging. We first trained four classic CNNs (InceptionV3[35], ResNet50[36], DenseNet[37], and InceptionResNetV2[38]) and selected the most outperformed network for further analyses. All images were resized to a resolution of 299 × 299 pixels at the training and inference stages for unified processing. The Adam optimizer was used throughout the whole procedure of the training stage[39]. The initial learning rate was set to 0.0001, and the learning rate decreased by a factor of 10 for fine learning when the mean squared error on the validation set stopped decreasing for three epochs. All the parameters were initialized with the ImageNet weights[40]. We trained the models for 50 epochs with a batch size of 32 and chose the model with optimal performance on the tuning set for each mode of image.

### Interpretable visualization of DL models

A gradient-weighted class activation map (Grad-CAM) was used to enhance the interpretability of our DL age estimation models[41]. The Grad-CAM algorithm can produce a class-specific activation heatmap where each activation value represents the importance of classifying to that class. However, predicting the age of a person is a regression problem, and Grad-CAM failed to generate a heatmap directly. To overcome this, we employed a workaround. We normalized the age values to the range of 0–1 by dividing them by 100 because an individual's age generally ranges from 0 to 100, and there were no participants over 100 years old in our datasets. By mapping the target age values to the same range as a classification problem, Grad-CAM could be effectively employed to produce meaningful heatmaps for age estimation.

To further interpret the DL models, we evaluated the model prediction when masking different lens structures for the lens images for slit-lamp mode. Lens structures including cortex, nucleus, and capsule were separately masked for each lens image in the validation set. All the LensAge predictive results of each image with these three lens structures masked were used for analysis. Then, adjusted logistic regression models were used to compare the influence of masking the cortex on predictive errors between cortical cataracts and other cataracts, the influence of masking the nucleus on predictive errors between nuclear cataracts and other cataracts, and the influence of masking the capsule on predictive errors between subcapsular cataracts and other cataracts.

### Measuring an individual's aging level

LensAge at the individual level for diffuse-light or slit-lamp mode was calculated by averaging the LensAge values generated by the DL models of all diffuse-light or slit-lamp images corresponding to one individual. The difference between LensAge at the individual level and chronological age was used to unveil an individual's aging level and served as the LensAge index. A LensAge index above 0 indicated a higher level of aging than their peers of the same chronological age, while a LensAge index below 0 indicated a lower level of aging (Fig. 1b).

### Evaluation of the LensAge index in reflecting the risks of age-related diseases

A total of 3433 participants in the general population (mean age [± s.d.] of 66.0 [ ± 11.5] years) aged 20 to 96 years with available physical data were included as the analysis dataset to investigate the ability of the LensAge index to reflect age-related disease risks. The validation dataset for the DL models was also included in the analysis dataset.

The ability of the LensAge index to reflect the risks of ocular age-related conditions, including moderate or severe visual impairment (visual acuity [VA] of the better eye less than 0.3), senile cataracts, and vitreous opacity, was assessed with adjusted ORs using logistic regression models. The ability of the LensAge index to reveal the risks of systemic age-related diseases, including age-related chronic diseases (diabetes, hypertension, coronary heart disease, cancer, and cerebral infarction), age-related chest X-ray findings (arteriosclerosis and left ventricular hypertrophy), and age-related electrocardiograph findings (myocardial ischemia, myocardial infarction, atrial fibrillation, and hypertensive heart disease), was also assessed with adjusted ORs using logistic regression models. The association of the LensAge index with BG level was investigated with the $\beta$ value using the adjusted linear regression models. The individuals were grouped into < 60 and ≥ 60 years for further analysis. Furthermore, the AUCs of the LensAge index and chronological age for predicting the occurrence of age-related diseases among the participants with a LensAge index in the lowest quartile or the highest quartile were determined and compared using paired DeLong tests.

### Evaluation of the predictive performance of the LensAge index for all-cause mortality

The participants in the analysis dataset were followed up from the time when the lens photographs were taken. To gather information on

all-cause mortality status and date of death, questionnaires were administered by the investigators to the relatives of the participants. The duration of follow-up for each participant was calculated as the time elapsed between their baseline and the date of death or the completion of the follow-up period (July, 2023), whichever came first. The percentage of individuals in the analysis dataset lost to follow-up was 13.1%. After a median follow-up of 30.2 months (IQR 28.0–32.0 months), a total of 66 (2.2%) individuals died from all causes. The baseline information of both individuals who were followed up and those who were lost to follow-up was comparable (Supplementary Table 15). To assess the predictive performance of the LensAge index for all-cause mortality risk, Cox proportional hazards regression models adjusted for chronological age, sex, race, region, occupation, smoking, and alcohol intake status were utilized. These models estimated the impact of a 1-year increase in the LensAge index on the risk of all-cause mortality. Additionally, we compared all-cause mortality of participants in the different quartiles of the LensAge index with those in the lowest quartile for reference.

### Lens aging assessment using smartphones
We further assessed whether our DL-based age estimation model based on smartphone photographs can evaluate biological age. Non-ophthalmologist volunteers used their own smartphones attached to a portable slit lamp to assist patients in capturing lens photographs according to our instructions (Supplementary Fig. 2). A total of 389 qualified lens photographs (accounting for 94.4% of all photographs taken) from 102 individuals (mean age [± s.d.] of 62.0 [± 10.8] years) were obtained using iPhone/Huawei smartphones attached to a portable slit lamp. Among these, 157 images from 50 participants (mean age [± s.d.] of 64.6 [± 11.4] years) without a medical history of diseases were used as a reference dataset of relatively healthy individuals for accuracy estimation of the DL model of slit-lamp mode. Furthermore, the ability of the LensAge index to reflect the risks of age-related chronic diseases (diabetes, hypertension, coronary heart disease, cancer, and cerebral infarction) based on smartphone photographs taken in slit-lamp mode was investigated with adjusted ORs using logistic regression models.

### Statistical analysis
The MAEs at both the image level and the individual level and $R^2$ at the individual level were used to evaluate the performance of the DL-based age estimation models. Adjusted logistic regression models were used to analyze the influence of masking different lens structures on the model prediction and the ability of the LensAge index to reflect age-related disease risks. The results are reported as adjusted ORs. Adjusted linear regression models were used to investigate the association of the LensAge index with BG level, and the results are reported as the adjusted $\beta$. Adjusted Cox proportional hazards regression models were used to evaluate the association between the LensAge index with all-cause mortality and the results are reported as the adjusted HRs. All regression models were adjusted for chronological age, sex, race, region, occupation, smoking, and alcohol intake status. Paired DeLong tests were used to compare the AUCs of the LensAge index and chronological age for predicting the occurrence of age-related diseases. Descriptive statistics, including means and s.ds., numbers and percentages, were used to report the baseline characteristics of the participants. Subgroup analyses stratified by sex and age group (20–40, 40–60, and ≥ 60 years) were performed. A two-sided $p$-value of < 0.05 indicated statistical significance. All statistical analyses were performed using R Statistics (version 4.1.2) or SPSS (version 20.0). Plots were created using the ggplot2 package (version 3.3.5) for R Statistics.

### Reporting summary
Further information on research design is available in the Nature Portfolio Reporting Summary linked to this article.

## Data availability
All data supporting the findings described in this manuscript are available in the article and in the Supplementary Information or/and from the corresponding author upon request. Data used to generate the main and supplementary figures are provided in the Source Data file. The patient data used in this study cannot be shared publicly due to privacy restrictions. However, in the case of noncommercial use, researchers can sign the license, complete a data access form provided at Github [https://github.com/RYL-gif/LensAge] and contact H.L. [linht5@mail.sysu.edu.cn] to access the de-identified representative images. For requests from verified academic researchers, access will be evaluated by the data access committee and be granted within one month. Source data are provided with this paper.

## Code availability
The code for the development of the LensAge index is available for academic and noncommercial use. Researchers can sign the license, complete a code access form provided at Github [https://github.com/RYL-gif/LensAge] and contact H.L. [linht5@mail.sysu.edu.cn] to access the code. For requests from verified academic researchers, access will be evaluated by the code access committee and be granted within one month.

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

## Acknowledgements

This study was funded by the National Natural Science Foundation of China (82171035 to H.L., 82150710555 and 82220108016 to X.L.), the Science and Technology Planning Projects of Guangdong Province (2021B1111610006 to H.L.), the Key-Area Research and Development of Guangdong Province (2020B1111190001 to H.L.), China Postdoctoral Science Foundation (2022M713589 to W.C.), the Fundamental Research Funds of the State Key Laboratory of Ophthalmology (2022QN10 to W.C.), Fundamental Research Funds for the Central Universities, Sun Yat-sen University (23ptpy120 to W.C.), the Program of Guangzhou Scientific Research Plan (202102010179 to X.L.), and Hainan Province Clinical Medical Center (H.L.). The funders had no role in the study design, data collection and analysis, decision to publish, or preparation of the manuscript.

## Author contributions

H.L., R.L., and W.C. contributed to the concept of the study and designed the research; R.L., R.W., Y.L., X.C., X.T, D.L., X.W., Y.S., A.X., X.W., D.W., X.Z., M.D., and Y.H. collected the data; R.L., W.C., and M.L. conducted the study; R.L., W.C., and L.Z. analyzed the data; H.L., R.L., W.C., and M.L. cowrote the manuscript; C.C., Y.Z., X.L., C.L., Y.H., L.Z., H.O., and M.L. critically revised the manuscript; H.L. and Z.L. performed the technical review; and all authors discussed the results and provided comments regarding the manuscript.

## Competing interests

The authors declare no competing interests.
