## [Peer Review File · Nature Communications]

LensAge index as a deep learning-based biological age for self-monitoring the risks of age-related diseases and mortalityREVIEWER COMMENTS

Reviewer #1 (Remarks to the Author):

The main contribution of this paper is the development of deep learning (DL) system to assess LensAge (and its derivative LensAge index) based on ocular anterior segment photograph. LensAge serves as a non-invasive method to assess biological age based on lens changes, to provide a quantitative indicator of the risks of age-related diseases.

Overall, the research question of this paper is novel. It demonstrated that the assessment of biological age using LensAge and its derivative LensAge index were accurate indicators of occurrence of age-related ocular (moderate/severe visual impairment, senile cataracts, vitreous opacity) and systemic (diabetes mellitus, hypertension, coronary heart disease, cancer, cerebral infarction; Chest X-ray findings (arteriosclerosis, left ventricular hypertrophy); ECG findings (Myocardial ischemia, infarction, atrial fibrillation, hypertensive heart disease); Increase in blood glucose levels) diseases.

(1) Major comments:

A. In terms of methodology:

The DL models for age estimation (LensAge) was trained on a large dataset of 8255 lens photographs (diffuse and slit-lamp) from 1990 relatively healthy individuals, with a relatively even distribution over a diverse age range from 20 to 96 years old. 4 CNNs (InceptionV3, ResNet50, DenseNet, InceptionResNetV2) were trained, and the most outperformed network (InceptionV3) was selected for further analyses.

- A1. All datasets consisted of participants from China. The reference dataset for training the DL models for age estimation (LensAge) consisted of participants recruited from a single national Chinese cataract screening program. The analysis dataset consisted of participants from a single hospital (Zhongshan Ophthalmic Center (Sun Yat-sen University, Guangdong, China)). The authors should include the breakdown of the participants' ethnicities in the list of baseline characteristics of the datasets (Table 1). The current LensAge DL model may be limited in generalizability to other populations with different ethnicities. This was acknowledged in the limitations discussed in the study.

- A2. In terms of inclusion/exclusion criteria for the reference dataset, what was defined as 'healthy individuals'? What is the full list of "medical history of diseases" and "ophthalmic examinations" that were recorded at baseline? For systemic health, were patients screened for diseases, or were patient-reported absence of diseases in questionnaires sufficient to qualify as 'healthy'? How did this account for patients who were unaware of underlying undiagnosed diseases?

- A3. It was noted that authors excluded participants who had previous eye surgery, trauma or ocular diseases. This was a good consideration as these may predispose to secondary cataracts that may be a

confounder for LensAge assessment. Were patients who were on long-term corticosteroids excluded as well?

- A4. A variety of slit-lamps were used to capture diffuse-light and slit-lamp anterior segment photographs including BQ-900, BX-900, OVS-II, and PSL-Classic. Did every participant have photographs captured using all slit lamps? Otherwise, what was the distribution of slit-lamp models used in the dataset?

B. In terms of explainability:

- B1. The authors used GradCAM attention maps to represent the LensAge DL model's attention areas, and concluded based on qualitative assessment of attention maps that the model focused on corresponding lens areas in the anterior segment photographs. Were GradCAM attention maps similar for participants with different colored iris pigmentation, irregular pupil, or corneal opacities?

- B2. Were there differences in prediction of LensAge or attention maps for different types of cataracts (Eg. nuclear sclerotic (Seen in Figure. 2e and f), cortical, posterior subcapsular, anterior subcapsular)?

C. In terms of clinical use case:

The authors further demonstrated that LensAge could be used to assess biological age using smartphone photographs with sufficient accuracy. This convenient self-monitoring method has the potential to widen the target population for large-scale implementation of LensAge.

- C1. In this study, smartphone photographs were captured using portable slit lamp (MediWorks portable slit lamp S150, Shanghai) attached to an iPhone/Huawei smart phone. Based on a brief Google search, MediWorks portable slit lamp S150 costs approximately USD\$500-1000. Were there cheaper alternatives explored that were still able to capture photographs of sufficient quality? Otherwise, this may limit the feasibility of large scale adoption of LensAge for the purposes of self-monitoring.

- C2. Were the smartphone photographs taken by an ophthalmologist/trained personnel? From the perspective of self-monitoring, will it be feasible for the patient to capture smartphone anterior segment photographs of sufficient quality for further analysis? Furthermore, it is unlikely possible to take the photograph him/herself, and will require another person like a caregiver to assist. Were there any specific lighting conditions required such as an ambient light source?

Reviewer #2 (Remarks to the Author):

The manuscript by Li et al. describes the development of a new measure of biological age that they call LensAge. They use two types of measurements to characterize the human eye lens: diffuse light mode and slit lamp mode. The images generated, on average about four per subject, are labeled with the

chronological age of the subject, and then subjected to deep learning with four different convolutional neural network (CNN) transfer learning algorithms. The highest performing algorithm based on its ability to classify subjects according to calendar age across the training set was used for further study. The studies included further assessments of the algorithm's performance characteristics, their ability to correlate with several age-related diseases, and utility for field use with a smartphone to secure images of a subject's eye lens.

The sort of approach described in this manuscript has become popular for classifying people by so-called biological age, in recent years. These approaches have focused on whole-body functional parameters, blood analytes, and intracellular molecules. They have also taken a similar approach to many different individual tissues. Furthermore, different analytic methods have been used to select individual features that ultimately become part of an algorithm that can predict all-cause mortality or specific disease risks. In only a few cases, have the studies focused only on the time since birth or calendar age of the subjects, as this manuscript does. Apart from the first-generation DNA methylation clocks, all of these approaches have proven to be successful to one extent or another. Interestingly, there appears to be little overlap between them in the portion of the variability in biological age that is explained. Biological age is almost always defined as the departure from the population mean, positive or negative.

Sample sizes, study design, and analytic methods implemented by Li et al. seem entirely appropriate to the stated tasks. Quite appropriately, the authors rest the novelty of their approach on the application of their lens measurement to a smartphone app that can be used outside a laboratory or clinical setting, although the illumination that is required goes beyond a simple smartphone, requiring an attached slit lamp.

The excellent correlation of LensAge with chronological age is not at all surprising, given that the latter is incorporated into the process of determining the former. It is surprising that this is stressed as a beneficial factor because it is the departure from the perfect correlation that is used to define the measure of biological age, LensAge Index. A surprising finding is that this LensAge Index becomes smaller and smaller with calendar age. This is not explained adequately. At face value, it suggests that at older ages LensAge Index becomes less and less useful as a metric. (This is also seen in most cases of disease risk, where the risk becomes lower in the very broad age group of >60.) The authors should address another question: What proportion of the LensAge Index (that is the departure from the population mean) is biological as opposed to technical? This could greatly affect conclusions drawn from the ROC curves.

The best predictor of age-related disease, including cataracts, is chronological age. Thus, the excellent fit between LensAge and chronological age is not at all surprising. The proteins of the lens turn over more slowly than any other tissue. Thus, they accumulate increasing damage over time, reflecting the passage of time very accurately. Might the LensAge Index, therefore, reflect largely passage of time and not some biological aging process as such.

A very useful feature of the LensAge Index is its potential ability to uncover eye disease risk, and perhaps risk of other diseases, fairly early in life, providing substantial clinical value. However, it would be important to know the differences in risk for individuals with a positive LensAge Index compared to the population mean and for individuals with a negative LensAge Index, separately. For example, comparing what is in the text of the Results section with Fig. 4d itself, it is unclear what the conclusion really is. The authors need to explain this figure better.

Reviewer #3 (Remarks to the Author):

This study developed a deep learning-based method to predict biological age based on the human lens. The deep learning model is developed on a dataset of 1,990 relatively healthy individuals. Then the model is used to predict the LensAge in the general population including 3,433 participants, where experiments showed that LensAge can reveal the risks of age-related diseases. The method is also validated on 389 smartphone images.

Strengths:

- 1) The research is comprehensive, including the experiment on various age-related diseases and the experiment of different methods.
- 2) The large number and wide types of subjects in this study make the study of high value and credibility.

Major Issues:

- 1) The key rationale for using lens as an indicator of age is not clear. The study expects LensAge to be an objective evaluation method for biological age (line 70). But the interpretation (line 129) of the DL model is not sufficient for it to be an objective method. In line 75, the lens can provide objective age-dependent changes including nucleus enlargement, elasticity reduction, and increased opacity. However, in line 133, the interpretation of the DL model is shown by heatmaps indicating that the DL models focus on the lens. Whether the DL model predicts the biological age based on the key factors of nucleus enlargement, elasticity reduction, and increased opacity still needs to be further discussed. To claim the objectiveness of LensAge, additional experiments are required.
- 2) What does LensAge measure? The authors should clarify what is used as ground truth for training the DL model to predict lens ages. If the ground truth is biological age, the authors should specify how the biological age is calculated. If the ground truth is chronological age during the training phase, a concern is whether the DL model truly learns the biological age or just simply approximates the chronological age. In line 653, the AUC for LensAge and chronological age are close. The authors can consider collecting a precise biological age as described in lines 63-69 for evaluating the precision of the biological

age predicted by the DL model. The authors can also consider comparing the LensAge and the chronological age on a group of diseased people to see if chronological age cannot accurately reveal the risks of age-related diseases in this scenario.

3) There is some lack of consistency in results in some analysis. In Table 4-7, some p-values > 0.1, so the effectiveness of LensAge in these “negative associations” should be further discussed.

4) Novelty of research unclear. There has been some research [1,2] to predict biological age from retinal imaging. The authors should make proper comparisons (quantitative/qualitative) and citations, and state the novelty clearly of LensAge over the previous works.

5) Impact of research unclear. The authors suggest smart phone cameras can measure LensAge. However, whether this is clinically useful is not demonstrated.

6) Lack of validation in external dataset. There are lens data from Singapore Epidemiology Studies and the also the US AREDS dataset. It is unclear how applicable is this algorithm outside of Chinese population. The authors may wish to discuss this

Reference

[1] Liu, C. et al. Biological Age Estimated from Retinal Imaging: A Novel Biomarker of Aging. In: Medical Image Computing and Computer Assisted Intervention – MICCAI 2019. MICCAI 2019.

[2] Simon N. et al. Retinal photograph-based deep learning predicts biological age, and stratifies morbidity and mortality risk, Age and Ageing, 2022

Re: “LensAge: A deep learning-based biological age for self-monitoring disease risks” (NCOMMS-23-04543-T)

Reviewers’ Comments:

Reviewer #1 (Remarks to the Author):

The main contribution of this paper is the development of deep learning (DL) system to assess LensAge (and its derivative LensAge index) based on ocular anterior segment photograph. LensAge serves as a non-invasive method to assess biological age based on lens changes, to provide a quantitative indicator of the risks of age-related diseases. Overall, the research question of this paper is novel. It demonstrated that the assessment of biological age using LensAge and its derivative LensAge index were accurate indicators of occurrence of age-related ocular (moderate/severe visual impairment, senile cataracts, vitreous opacity) and systemic (diabetes mellitus, hypertension, coronary heart disease, cancer, cerebral infarction; Chest X-ray findings (arteriosclerosis, left ventricular hypertrophy); ECG findings (Myocardial ischemia, infarction, atrial fibrillation, hypertensive heart disease); Increase in blood glucose levels) diseases.

Response:

We thank the Reviewer for the positive feedback on the importance and novelty of our work.

(1) Major comments:

A. In terms of methodology:

The DL models for age estimation (LensAge) was trained on a large dataset of 8255 lens photographs (diffuse and slit-lamp) from 1990 relatively healthy individuals, with a relatively even distribution over a diverse age range from 20 to 96 years old. 4 CNNs (InceptionV3, ResNet50, DenseNet, InceptionResNetV2) were trained, and the most outperformed network (InceptionV3) was selected for further analyses.

- A1. All datasets consisted of participants from China. The reference dataset for training the DL models for age estimation (LensAge) consisted of participants recruited from a single national Chinese cataract screening program. The analysis dataset consisted of participants from a single hospital (Zhongshan Ophthalmic Center (Sun Yat-sen University, Guangdong, China)). The authors should include the breakdown of the participants' ethnicities in the list of baseline characteristics of the datasets (Table 1). The current LensAge DL model may be limited in generalizability to other populations with different ethnicities. This was acknowledged in the limitations discussed in the study.

Response:

We have provided a breakdown of the participants' nationalities in the list of baseline characteristics in **Table 1** in the revised manuscript as requested by the Reviewer. Although most of the participants enrolled in this study were Chinese, a number of participants from other nationalities were included in our datasets. We have provided the performance of our models in this non-Chinese population to show the potential generalizability of our LensAge models across ethnicities and nationalities (**Supplementary Fig. 1**). In addition, we used a smartphone dataset from other centers to investigate the effectiveness and generalizability of the LensAge models. Our models were shown to demonstrate a certain level of generalizability to other external datasets. In the current study, our primary goal was to evaluate the feasibility of evaluating biological age based on age-related features in lens for various applications and self-monitoring. In the future, an important continuation of this work will expand the study population to other ethnic groups and countries. We have included this point in the limitation section (**Page 14 Line 369-373**):

“Second, although a small proportion of non-Chinese people were included in our study, showing the potential generalizability of our LensAge models across additional ethnicities and nationalities, further larger-scale validation to other ethnic groups and countries is needed as an important continuation of this work.”

Table 1 | Baseline characteristics of the datasets

	Traditional slit-lamp images		Smartphone images	
	Reference dataset	Analysis dataset	Reference dataset	Analysis dataset
No. of participants	1,990	3,433	50	102
Nationality, n (%)				
Chinese	1,952 (98.1%)	3,370 (98.2%)	50 (100%)	99 (97.1%)
Non-Chinese	38 (1.9%)	63 (1.8%)	0	3 (2.9%)
Age in years (mean±s.d.)	55.3±18.0	66.0±11.5	64.6±11.4	62.0±10.8
Distribution of chronological age, n (%)				
?20 and <30	245 (12.3%)	8 (0.2%)	0	0
?30 and <40	231 (11.6%)	55 (1.6%)	3 (6.0%)	4 (3.9%)
?40 and <50	246 (12.4%)	224 (6.5%)	0	6 (5.9%)
?50 and <60	292 (14.7%)	624 (18.2%)	15 (30.0%)	37 (36.3%)
?60 and <70	507 (25.5%)	1,139 (33.2%)	12 (24.0%)	28 (27.5%)
?70 and <80	334 (16.8%)	976 (28.4%)	18 (36.0%)	23 (22.5%)
?80	134 (6.7%)	407 (11.9%)	2 (4.0%)	4 (3.9%)
Sex, n (%)				
Male	732 (36.8%)	1,482 (43.2%)	20 (40.0%)	45 (44.1%)
Female	1,258 (63.2%)	1,951 (56.8%)	30 (60.0%)	57 (55.9%)
Images, n				
Diffuse-light images	4,542	5,641	N/A	N/A
Slit-lamp images	3,713	5,663	157	389

Supplementary Figure 1. The LensAge among the non-Chinese population. a, Scatterplot shows the correlation of LensAge at the individual level with chronological age for relatively healthy participants for diffuse-light mode ($p=5.28e-6$, two-sided linear regression, $n=20$). b, Scatterplot shows the correlation of LensAge at the individual level with chronological age for relatively healthy participants for slit-lamp mode ($p=1.17e-3$, two-sided linear regression, $n=19$).

- A2. In terms of inclusion/exclusion criteria for the reference dataset, what was defined as ‘healthy individuals’? What is the full list of “medical history of diseases” and “ophthalmic examinations” that were recorded at baseline? For systemic health, were patients screened for diseases, or were patient-reported absence of diseases in questionnaires sufficient to qualify as ‘healthy’? How did this account for patients who were unaware of underlying undiagnosed diseases?

Response:

“Relatively healthy” participants were defined as those who self-reported the absence of systemic medical conditions and had no abnormalities on regular physical examinations (including heart rate, blood pressure, respiratory rate, height, and weight) and ancillary examinations (including routine blood test, EEG, and chest X-ray). (Page 15 Line 397-404). We have clarified this definition in the Methods section in the revised manuscript (Page 16 Line 436-437):

“Relatively healthy participants were defined as those who did not report any medical

history of systemic diseases and had no physical examination abnormalities at baseline.” The full list of “medical history of diseases” included diabetes, hypertension, cardiovascular disease, cerebrovascular disease, cancer, and other chronic systemic diseases (**Page 15 Line 407-409**).

The full list of “ophthalmic examinations” consisted of functional and structural examinations, including visual acuity, intraocular pressure, slit-lamp examinations, funduscopy examinations, and cycloplegic refraction (**Page 15 Line 411-413**).

Indeed, we did not include 100% of patients with an ideal healthy status, and we used the term “relatively healthy”. We excluded most unhealthy patients based on medical history and physical examinations. Although some participants who were unaware of any underlying and undiagnosed diseases may have been included, we included a sufficient sample size of relatively healthy participants of different ages for DL model development to minimize the influence of unhealthy patients, adopting a similar approach employed in previous studies focusing on biological age assessments for model training [1,2]. In this way, our models can largely learn the average aging characteristics as reference among populations. In addition, we have added the following statement in the limitations section (**Page 14 Line 377-382**):

“Fourth, although we may include participants who were unaware of their underlying and undiagnosed diseases in the relatively healthy datasets, we excluded most unhealthy patients based on the medical history and physical examinations, and a sufficient sample size of relatively healthy participants of different ages was included for DL model development to minimize the influence of any patients with underlying conditions on models to learn the average aging characteristics among the population.”

References:

- [1] Zhu Z, Shi D, Guankai P, et al. Retinal age gap as a predictive biomarker for mortality risk[J]. *British Journal of Ophthalmology*, 2023, 107(4): 547-554.
- [2] Tian Y E, Cropley V, Maier A B, et al. Heterogeneous aging across multiple organ systems and prediction of chronic disease and mortality[J]. *Nature Medicine*, 2023: 111.

- A3. It was noted that authors excluded participants who had previous eye surgery, trauma or ocular diseases. This was a good consideration as these may predispose to secondary cataracts that may be a confounder for LensAge assessment. Were patients who were on long-term corticosteroids excluded as well?

Response:

Thank you for bringing this important point to our attention. We did not include patients with long-term corticosteroid use, which may cause drug-induced cataracts. This point may not have been clear in the original exclusion criteria, and so we have clarified this point in the revised manuscript (**Page 15 Line 404-407**):

“All enrolled participants were eligible for the study if they had no history of previous eye surgery, eye trauma, ocular diseases (high myopia, etc.), and long-term use of corticosteroids or other drugs that can cause complicated or drug-induced cataracts.”

- A4. A variety of slit-lamps were used to capture diffuse-light and slit-lamp anterior segment photographs including BQ-900, BX-900, OVS-II, and PSL-Classic. Did every participant have photographs captured using all slit lamps? Otherwise, what was the distribution of slit-lamp models used in the dataset?

Response:

Each participant had his/her photographs captured using one of these slit lamps. Because the images in reference dataset were randomly split into the training set, tuning set, and validation set for AI model development, the images from different slit-lamp models were evenly distributed among the three datasets. In addition, we have included information of proportions of slit-lamp models in the reference and analysis datasets in **Supplementary Table 13**.

Supplementary Table 13 | Distribution of images captured by different traditional slit lamps in the datasets

	Reference dataset	Analysis dataset
Diffuse-light images, n (%)	4,542 (100%)	5,641 (100%)
BQ-900	1,339 (29.5%)	1,990 (35.3%)
BX-900	1,231 (27.1%)	1,074 (19.0%)
OVS-II	1,013 (22.3%)	1,009 (17.9%)
PSL-Classic	959 (21.1%)	1,568 (27.8%)
Slit-lamp images, n (%)	3,713 (100%)	5,663 (100%)
BQ-900	1,089 (29.3%)	1,689 (29.8%)
BX-900	1,101 (29.7%)	1,741 (30.7%)
OVS-II	993 (26.7%)	1,011 (17.9%)
PSL-Classic	530 (14.3%)	1,222 (21.6%)

B. In terms of explainability:

- B1. The authors used GradCAM attention maps to represent the LensAge DL model's attention areas, and concluded based on qualitative assessment of attention maps that the model focused on corresponding lens areas in the anterior segment photographs. Were GradCAM attention maps similar for participants with different colored iris pigmentation, irregular pupil, or corneal opacities?

Response:

As requested by the Reviewer, we have provided the specific GradCAM attention maps for irregular pupils and corneal opacities for review. In addition, because most of the participants in this study were Chinese with similar iris pigmentations, we conducted an additional trial to simulate different iris pigmentations to investigate their influence on the attention maps for review. The results indicate that our models are robust in focusing on the lens areas for decision-making.

Heatmap for cases with corneal opacities

Heatmap for cases with irregular pupils

Heatmap for cases with simulated different colored iris pigmentations

Figure for review. Heatmaps for LensAge prediction for different ocular characteristics.

- B2. Were there differences in prediction of LensAge or attention maps for different types of cataracts (Eg. nuclear sclerotic (Seen in Figure. 2e and f), cortical, posterior subcapsular, anterior subcapsular)?

Response:

Thanks for bringing this interesting point to our attention. Because GradCAM attention maps may not be sufficiently precise to highlight different structures of the lenses and have difficulty in quantitatively analyzing the influence of different forms of cataracts, we instead analyzed the prediction results of LensAge for different types of cataracts and investigated the influence of masking different lens structures on the model prediction for slit-lamp mode. Our findings demonstrate that our model effectively directed its attention towards the specific region associated with the corresponding

cataract type for decision-making. The updated results have been provided in the revised manuscript:

Results section-Page 5 Line 123-124

“Moreover, our models demonstrated reasonable performance among different types of cataracts (Supplementary Table 2).”

Results section-Page 6 Line 141-151

“To further increase the interpretability of the DL models, we analyzed the influence of masking different lens structures on LensAge prediction for slit-lamp mode. When masking the lens cortex for cortical and noncortical cataracts, the predictive error was higher among cortical cataracts (adjusted odds ratio [OR]=1.23, 95% confidence interval [CI] 1.16-1.31, $p=6.72e-11$, Supplementary Table 4), indicating that for this condition, the lens cortex regions were more important for model decision-making than for other types of cataracts. Similarly, the predictive errors were higher among nuclear cataracts and subcapsular cataracts than among other types of cataracts when masking the lens nucleus and lens capsule, respectively (lens nucleus, adjusted OR=1.24, 95% CI 1.16-1.33, $p=1.55e-10$, lens capsule, adjusted OR=1.12, 95% CI 1.02-1.22, $p=1.60e-2$, Supplementary Table 4).”

Supplementary Table 2 | LensAge prediction for different types of cataracts

LensAge mode	Cortical cataracts			Nuclear cataracts			Subcapsular cataracts		
	MAE	R ²	p value	MAE	R ²	p value	MAE	R ²	p value
Diffuse-light mode	4.66	0.68	<1.00e-36	4.78	0.64	<1.00e-36	4.51	0.73	4.68e-21
Slit-lamp mode	4.54	0.73	<1.00e-36	5.90	0.58	<1.00e-36	4.38	0.80	5.01e-29

MAE, mean absolute error. *p* value from a two-sided test using linear regression. **p* value<0.05.

Supplementary Table 4 | Analyses of the importance of different lens structures on LensAge prediction for different types of cataracts

Masking structure	MAE			Adjusted OR (95% CI)	p value
	Cortical cataract	Nuclear cataract	Subcapsular cataract		
Lens cortex	8.47	5.44	4.10	1.23 (1.16-1.31)	6.72e-11*
Lens nucleus	5.95	11.43	4.86	1.24 (1.16-1.33)	1.55e-10*
Lens capsule	5.61	4.25	5.84	1.12 (1.02-1.22)	1.60e-2*

Adjusted logistic regression models were used to compare the influence of masking the cortex on predictive errors between cortical cataracts and other cataracts, the influence of masking the nucleus on predictive errors between nuclear cataracts and other cataracts, and the influence of masking the capsule on predictive errors between subcapsular cataracts and other cataracts. The results are reported with adjusted ORs. MAE, mean absolute error; OR, odds ratio; CI, confidence interval. *p* value from a two-sided test using adjusted logistic regression. **p* value<0.001.

C. In terms of clinical use case:

The authors further demonstrated that LensAge could be used to assess biological age using smartphone photographs with sufficient accuracy. This convenient self-monitoring method has the potential to widen the target population for large-scale implementation of LensAge.

- C1. In this study, smartphone photographs were captured using portable slit lamp (MediWorks portable slit lamp S150, Shanghai) attached to an iPhone/Huawei smart phone. Based on a brief Google search, MediWorks portable slit lamp S150 costs approximately USD\$500-1000. Were there cheaper alternatives explored that were still able to capture photographs of sufficient quality? Otherwise, this may limit the feasibility of large scale adoption of LensAge for the purposes of self-monitoring.

Response:

We thank the reviewer for bringing this point to our attention. To measure the biological age of individuals is of great importance but is limited by a lack of objective, reliable, convenient, and noninvasive assessment methods. An important aim of the current feasibility study was to design and validate a feasible smartphone-based technique for

the convenient and noninvasive assessment of biological age and for self-monitoring, which will promote the further application of biological age assessment. Although at this stage, the price advantage of the portable slit lamp we used is not particularly obvious, our findings indicate that the smartphone-based method for biological age assessment is feasible and promising. In the future, more engineering efforts will be made as a major part of our work to apply less expensive alternatives to LensAge assessment for broader applicability. We have added a statement in the Discussion section in the revised manuscript (**Page 12 Line 330-337** and **Page 13 Line 338-344**): *“More importantly, lens features can be conveniently and objectively captured by portable devices to identify disease states. Previous assessments of biological age mostly focused on technologies that were not convenient for or available to patients outside the hospital; in other words, they could not be implemented in the community or at home. We applied our smartphone-based method to assess biological age, showing great potential for self-monitoring aging status when needed. Given the rapid development of mobile health (mHealth) and the application of mobile devices, medical conditions can be monitored and intervened whenever and wherever. LensAge based on smartphone photographs facilitates routine self-assessment of disease risks and aging/health status. Although our smartphone-based LensAge assessment currently still requires attaching a portable slit lamp for capturing lens photographs, the mHealth mode for the self-monitoring of aging status was shown to be effective. In the future, a large-scale prospective trial is needed to validate the utility of the smartphone-based determination of LensAge in real-world applications, and a more convenient or inexpensive alternative to portable slit lamps must be explored.”*

- C2. Were the smartphone photographs taken by an ophthalmologist/trained personnel? From the perspective of self-monitoring, will it be feasible for the patient to capture smartphone anterior segment photographs of sufficient quality for further analysis? Furthermore, it is unlikely possible to take the photograph him/herself, and will require another person like a caregiver to assist. Were there any specific lighting conditions required such as an ambient light source?

Response:

Thank you for bringing this important point to our attention. In our study, nonophthalmologist volunteers used their own smartphones to assist patients in capturing lens photographs according to our standard instructions, and importantly, this process is convenient and reliable for further applications in community and at-home settings, with 94.4% images qualified to be included in the analysis. We have provided an illustration of the brief instructions for capturing lens photographs using smartphones attached to a portable slit lamp (**Supplementary Fig. 2**). There are no requirements for specific ambient lighting sources in addition to the light source provided by the portable slit lamp and camera of smartphones, and normal daily ambient lighting conditions are sufficient, provided they are not overexposed. We have added a clearer statement of this point in the Methods section (**Page 19 Line 516-518**): “*Nonophthalmologist volunteers used their own smartphones attached to a portable slit lamp to assist patients in capturing lens photographs according to our instructions (Supplementary Fig. 2).*”

	1. Choose an appropriate environment without direct light 	Overexposure 2. Open the smartphone camera and set the portable slit lamp with appropriate brightness and narrow slit 	Strong brightness or wide slit 3. Keep the smartphone in a stable position and appropriate distance from patient 	Deviated 4. Take the lens photographs when the lens area is clearly in focus 	Blurring, without lens area, et al. 
Supplementary Figure 2. Brief instructions for capturing lens photographs using smartphones.

Reviewer #2 (Remarks to the Author):

The manuscript by Li et al. describes the development of a new measure of biological age that they call LensAge. They use two types of measurements to characterize the human eye lens: diffuse light mode and slit lamp mode. The images generated, on average about four per subject, are labeled with the chronological age of the subject, and then subjected to deep learning with four different convolutional neural network (CNN) transfer learning algorithms. The highest performing algorithm based on its ability to classify subjects according to calendar age across the training set was used for further study. The studies included further assessments of the algorithm's performance characteristics, their ability to correlate with several age-related diseases, and utility for field use with a smartphone to secure images of a subject's eye lens.

The sort of approach described in this manuscript has become popular for classifying people by so-called biological age, in recent years. These approaches have focused on whole-body functional parameters, blood analytes, and intracellular molecules. They have also taken a similar approach to many different individual tissues. Furthermore, different analytic methods have been used to select individual features that ultimately become part of an algorithm that can predict all-cause mortality or specific disease risks. In only a few cases, have the studies focused only on the time since birth or calendar age of the subjects, as this manuscript does. Apart from the first-generation DNA methylation clocks, all of these approaches have proven to be successful to one extent or another. Interestingly, there appears to be little overlap between them in the portion of the variability in biological age that is explained. Biological age is almost always defined as the departure from the population mean, positive or negative.

Sample sizes, study design, and analytic methods implemented by Li et al. seem entirely appropriate to the stated tasks. Quite appropriately, the authors rest the novelty of their approach on the application of their lens measurement to a smartphone app that can be used outside a laboratory or clinical setting, although the illumination that is required goes beyond a simple smartphone, requiring an attached slit lamp.

Response:

We thank the Reviewer for the positive feedback on the importance and novelty of our work. We investigated the ability of LensAge to reflect biological age and age-related disease risks through several steps and did not solely focus on calendar age. First, we used lens photographs from a relatively **healthy population** with chronological age labels to train the deep learning models as chronological age predictors to learn the average aging characteristics for biological age reference. Chronological age predictors have shown considerable promise as proxies of biological age of individuals by learning the population norm in the context of aging [1,2]. Second, the difference between the “LensAge” generated by deep learning models and chronological age was used to unveil an individual’s aging level **among the general population** and served as the “LensAge index”. It has been widely hypothesized that the difference between the estimated and actual chronological age, called the ‘ Δ age’ or ‘age gap’, reflects variations in their prior rates of aging [2-4]. These hypotheses have been supported by observations that individuals with positive age gaps, termed age acceleration, are at greater risk of mortality and certain aging diseases, such as heart disease, metabolic disease, and cancers [5-7]. Therefore, in the third step of our study, we analyzed the “LensAge index” to reveal the risk of age-related diseases among the **general population**. Furthermore, we rested the novelty of our approach on the application of biological age measurement to a smartphone app that can be used outside the clinical setting.

References:

- [1]Galkin F, Mamoshina P, Aliper A, et al. Biohorology and biomarkers of aging: Current state-of-the-art, challenges and opportunities[J]. Ageing Research Reviews, 2020, 60: 101050.
- [2]Rutledge J, Oh H, Wyss-Coray T. Measuring biological age using omics data[J]. Nature Reviews Genetics, 2022: 1-13.
- [3]Hannum G, Guinney J, Zhao L, et al. Genome-wide methylation profiles reveal quantitative views of human aging rates[J]. Molecular Cell, 2013, 49(2): 359-367.

[4]Horvath S. DNA methylation age of human tissues and cell types[J]. Genome Biology, 2013, 14(10): 1-20.

[5]Chen B H, Marioni R E, Colicino E, et al. DNA methylation-based measures of biological age: meta-analysis predicting time to death[J]. Aging (Albany NY), 2016, 8(9): 1844.

[6]Tanaka T, Biancotto A, Moaddel R, et al. Plasma proteomic signature of age in healthy humans[J]. Aging Cell, 2018, 17(5): e12799.

[7]Levine M E, Lu A T, Quach A, et al. An epigenetic biomarker of aging for lifespan and healthspan[J]. Aging (albany NY), 2018, 10(4): 573.

The excellent correlation of LensAge with chronological age is not at all surprising, given that the latter is incorporated into the process of determining the former. It is surprising that this is stressed as a beneficial factor because it is the departure from the perfect correlation that is used to define the measure of biological age, LensAge Index. A surprising finding is that this LensAge Index becomes smaller and smaller with calendar age. This is not explained adequately. At face value, it suggests that at older ages LensAge Index becomes less and less useful as a metric. (This is also seen in most cases of disease risk, where the risk becomes lower in the very broad age group of >60.) The authors should address another question: What proportion of the LensAge Index (that is the departure from the population mean) is biological as opposed to technical? This could greatly affect conclusions drawn from the ROC curves.

Response:

As mentioned in the previous response to the first comment of the Reviewer, we investigated the ability of LensAge to reflect biological age and age-related disease risks through several steps. In the first step, we used lens photographs from a relatively healthy population to train the deep learning models as chronological age predictors. To evaluate the ability of the models to learn the **normative aging characteristics of the relatively healthy population**, we evaluated the correlation of LensAge with chronological age in a method similar to what have been applied for the accuracy

assessment of other biological age models in previous studies [1,2]. Furthermore, the difference between the “LensAge” and the true chronological age **in the general population**, termed as the “LensAge index”, was used to evaluate the departure of individuals’ lens aging paces from the norm derived from a relatively healthy population and to reveal age-related disease risks. It has been widely hypothesized that this estimated age can serve as a measure of an individual’s biological age and that the difference between the estimated and actual chronological age, called the ‘ Δ age’ or ‘age gap’, reflects variations in their prior rates of aging [3-4]. A positive LensAge index indicated a faster pace of aging and higher risks of age-related diseases among peers with the same chronological age.

Our results show that the LensAge index decreases for chronological ages ≥ 60 years, which can be attributed to the aging disparity that plateaus at older ages. Aging displays various paces among different age groups, with smaller aging disparity and steady states in chronological ages older than 60 years old reported in previous studies [2,5], implying that the aging interventions should be provided in the stage of large heterogeneity of aging rate. However, it does not diminish the clinical merit of LensAge in this age group. The biological age index reflects individual’s pace of aging relative to the same-age peers. Among the population ≥ 60 years old, a positive index was significantly correlated with a higher risk of age-related diseases (**Supplementary Table 6**), demonstrating that the LensAge index can effectively reveal disease risks in this age group.

Importantly, to investigate the ability of the LensAge index to reveal biological aging as requested by the Reviewer, we adjusted the regression models for chronological age as a covariate (Method section **Page 19 Line 537**). Subsequent to this adjustment, the LensAge index was still significantly correlated with an increased risk of age-related diseases (**Figure 3 a-b**) and changes in blood glucose (**Supplementary Table 12**). These findings indicate that LensAge can be an effective proxy of biological age and an independent risk marker for age-related diseases.

Furthermore, to show the clinical merit of LensAge, we added the ROC curve analysis of the LensAge index versus chronological age for predicting the occurrence of age-

related diseases among individuals with a LensAge index less than the 25th percentile or more than the 75th percentile, **showing wider AUC gaps**. The updated results have been provided in the revised manuscript (**Figure 3c-d** and **Page 9 Line 243-250** and **Page 10 Line 251-254**). **All the abovementioned results demonstrate that LensAge can be an effective proxy of biological age as opposed to technical artifacts.**

*“We further compared the ability of the LensAge index to predict the occurrence of age-related diseases with that of chronological age using receiver operating characteristic (ROC) curve analysis among the participants with a LensAge index less than the 25th percentile or more than the 75th percentile. The ROC curves graphically illustrate the predictive performance of the LensAge index with an area under the curve (AUC) of 0.621 (95% CI 0.596-0.645) for diffuse-light mode (**Fig. 3c**), and 0.600 (95% CI 0.575-0.624) for slit-lamp mode (**Fig. 3d**). Compared with those of chronological age, the AUCs of the LensAge index were significantly greater (diffuse-light mode, difference in AUCs=0.098, 95% CI 0.077-0.120, Z=8.968, $p<0.0001$, **Fig. 3c**; slit-lamp mode, difference in AUCs=0.090, 95% CI 0.030-0.151, Z=2.924, $p=3.50e-3$, **Fig. 3d**), demonstrating that LensAge can better reflect the aging process in humans and can be an optimized indicator of age-related disease risks.”*

Figure 3. The ability of LensAge to reveal age-related disease risks. a, Comparison of age-related changes between individuals of all ages with positive and negative LensAge indexes for diffuse-light mode or slit-lamp mode. *p* value from a two-sided test using adjusted logistic regression. **b,** Association of the LensAge index with age-related changes in individuals of all ages with a positive LensAge index for diffuse-light mode or slit-lamp mode. *p* value from a two-sided test using adjusted logistic regression. **c,** Comparison of the AUCs between the LensAge index and chronological age in predicting the occurrence of age-related diseases for diffuse-light mode among the participants with a LensAge index less than the 25th percentile or more than the 75th percentile. LensAge index, AUC=0.621 (95% CI 0.596 to 0.645); chronological age,

AUC=0.523 (95% CI 0.498 to 0.548); difference in AUCs=0.098 (95% CI 0.077 to 0.120), Z=8.968, p<0.0001, n=1,555, two-sided paired DeLong test. d, Comparison of the AUCs between the LensAge index and chronological age in predicting the occurrence of age-related diseases for slit-lamp mode among the participants with a LensAge index less than the 25th percentile or more than the 75th percentile. LensAge index, AUC=0.600 (95% CI 0.575 to 0.624); chronological age, AUC=0.509 (95% CI 0.484 to 0.535); difference in AUCs=0.090 (95% CI 0.030 to 0.151), Z=2.924, p=3.50e-3, n=1,536, two-sided paired DeLong test. CI, confidence interval; ROC, receiver operating characteristic; AUC, area under the curve.

References:

- [1]Rutledge J, Oh H, Wyss-Coray T. Measuring biological age using omics data[J]. Nature Reviews Genetics, 2022: 1-13.
- [2]Zhu Z, Shi D, Guankai P, et al. Retinal age gap as a predictive biomarker for mortality risk[J]. British Journal of Ophthalmology, 2023, 107(4): 547-554.
- [3]Hannum G, Guinney J, Zhao L, et al. Genome-wide methylation profiles reveal quantitative views of human aging rates[J]. Molecular Cell, 2013, 49(2): 359-367.
- [4]Horvath S. DNA methylation age of human tissues and cell types[J]. Genome Biology, 2013, 14(10): 1-20.
- [5]Xia X, Chen X, Wu G, et al. Three-dimensional facial-image analysis to predict heterogeneity of the human ageing rate and the impact of lifestyle[J]. Nature metabolism, 2020, 2(9): 946-957.

The best predictor of age-related disease, including cataracts, is chronological age. Thus, the excellent fit between LensAge and chronological age is not at all surprising. The proteins of the lens turn over more slowly than any other tissue. Thus, they accumulate increasing damage over time, reflecting the passage of time very accurately. Might the LensAge Index, therefore, reflect largely passage of time and not some biological aging process as such.

Response:

We thank the Reviewer for bringing this important issue to our attention. As we have stated in the previous responses to the Reviewer, we investigated the ability of LensAge to reflect biological age and age-related disease risks through several steps. We evaluated the correlation of LensAge with chronological age **among the relatively healthy population**, and the excellent fit between LensAge and chronological age indicated that the models learned the normative aging characteristics of the relatively healthy population, a method similar to what have been applied in the accuracy assessment of other biological age models in previous studies [1-3]. We agree that to a certain extent, it is a factor related to the accumulation of temporal age that causes age-related diseases. However, **among the general population**, we observed a substantial number of participants who showed a LensAge inconsistent with chronological age, reflecting their variations in prior rates of aging, which cannot be merely accounted for by the passage of time. One of the important clinical merit of many biological age markers is the ability to detect participants with aging processes that are different from the norm.

More importantly, to investigate the clinical merit of LensAge in revealing biological age, we adjusted the regression models for chronological age, and the LensAge index was also significantly correlated with an increased risk of age-related diseases and changes in blood glucose, indicating that LensAge can be an appropriate proxy of biological age. Furthermore, we analyzed the differences in risk of age-related diseases for individuals with a positive LensAge index compared with that for individuals with a negative LensAge index. We have also added the analysis of the differences in risk of age-related diseases between individuals with a LensAge index above the 75th percentile and individuals with a LensAge index at the moderate level (between the 25th and 75th percentiles). All regression models were adjusted for chronological age to investigate whether LensAge can be an effective marker for revealing biological age, independent of passage of time. The results are provided in the revised manuscript (**Figure 3a; Page 7 Line 191-192, Page 8 Line 193-195 and Line 215-221; Page 9 Line 222-227**):

“Compared with individuals with a negative LensAge index for diffuse-light mode, those with a positive LensAge index had a higher risk of moderate or severe visual impairment (adjusted OR=1.65, 95% CI 1.31-2.08, p=1.80e-5), senile cataracts (adjusted OR=1.76, 95% CI 1.45-2.14, p=1.50e-8), and vitreous opacity (adjusted OR=1.89, 95% CI 1.27-2.81, p=1.84e-3) in individuals of all ages (Fig. 3a).”

“Compared to those with a negative LensAge index for diffuse-light mode, those with a positive LensAge index had a higher risk of systemic age-related diseases (diabetes, hypertension, coronary heart disease, cancer, cerebral infarction) (adjusted OR=1.26, 95% CI 1.05-1.52, p=0.010), age-related changes in chest X-ray findings (arteriosclerosis and left ventricular hypertrophy) (adjusted OR=1.50, 95% CI 1.17-1.92, p=0.010), and age-related changes in electrocardiographic findings (myocardial ischemia, myocardial infarction, atrial fibrillation, and hypertensive heart disease) (adjusted OR=1.27, 95% CI 1.08-1.50, p=0.004) in individuals of all ages (Fig. 3a).” *“In addition, individuals with the LensAge index above the 75th percentile had a higher risk of age-related diseases than those with the LensAge index between the 25th and 75th percentiles (diffuse-light mode, adjusted OR=1.61, 95% CI 1.31-1.97, p=5.46e-6, slit-lamp mode, adjusted OR=1.23, 95% CI 1.00-1.53, p=4.73e-2, Supplementary Table II).”*

Moreover, we added the ROC analysis of the LensAge index versus chronological age for predicting the occurrence of age-related diseases among the participants with a LensAge index less than the 25th percentile or more than the 75th percentile to show the clinical merit of LensAge. The updated results have been provided in the revised manuscript (**Figure c-d** and **Page 9 Line 243-250** and **Page 10 Line 251-254**):

“We further compared the ability of the LensAge index to predict the occurrence of age-related diseases with that of chronological age using receiver operating characteristic (ROC) curve analysis among the participants with a LensAge index less than the 25th percentile or more than the 75th percentile. The ROC curves graphically illustrate the predictive performance of the LensAge index with an area under the curve (AUC) of

0.621 (95% CI 0.596-0.645) for diffuse-light mode (**Fig. 3c**), and 0.600 (95% CI 0.575-0.624) for slit-lamp mode (**Fig. 3d**). Compared with those of chronological age, the AUCs of the LensAge index were significantly greater (diffuse-light mode, difference in AUCs=0.098, 95% CI 0.077 to 0.120, Z=8.968, $p<0.0001$, **Fig. 3c**; slit-lamp mode, difference in AUCs=0.090, 95% CI 0.030 to 0.151, Z=2.924, $p=3.50e-3$, **Fig. 3d**), demonstrating that LensAge can better reflect the aging process in humans and can be an optimized indicator of age-related disease risks.”

Supplementary Table 11 | Risks of age-related changes for the LensAge index above the 75th percentile versus the LensAge index between the 25th and 75th percentiles

	Diffuse-light mode		Slit-lamp mode	
	Adjusted odds ratio (95% CI)	p value	Adjusted odds ratio (95% CI)	p value
Eye aging				
Moderate or severe visual impairment	3.00 (2.28-3.95)	3.67e-15*	2.32 (1.77-3.05)	1.18e-9*
Senile cataracts	2.61 (2.02-3.37)	1.54e-13*	1.35 (1.07-1.71)	1.29e-2*
Vitreous opacity	1.50 (0.98-2.28)	6.04e-2	1.57 (1.04-2.35)	3.07e-2*
Systemic aging				
Age-related chronic diseases	1.61 (1.31-1.97)	5.46e-6*	1.23 (1.00-1.53)	4.73e-2*
The age-related findings of chest X-ray	1.50 (1.15-1.94)	2.49e-3*	1.13 (0.87-1.47)	3.63e-1
The age-related findings of electrocardiograms	1.42 (1.18-1.72)	2.69e-4*	1.08 (0.89-1.30)	4.54e-1

CI=confidence interval. *p* value from a two-sided test using adjusted logistic regression. **p* value<0.05.

Reference:

- [1] Rutledge J, Oh H, Wyss-Coray T. Measuring biological age using omics data[J]. Nature Reviews Genetics, 2022: 1-13.
- [2] Hannum G, Guinney J, Zhao L, et al. Genome-wide methylation profiles reveal quantitative views of human aging rates[J]. Molecular Cell, 2013, 49(2): 359-367.

[3] Zhu Z, Shi D, Guankai P, et al. Retinal age gap as a predictive biomarker for mortality risk[J]. *British Journal of Ophthalmology*, 2023, 107(4): 547-554.

A very useful feature of the LensAge Index is its potential ability to uncover eye disease risk, and perhaps risk of other diseases, fairly early in life, providing substantial clinical value. However, it would be important to know the differences in risk for individuals with a positive LensAge Index compared to the population mean and for individuals with a negative LensAge Index, separately. For example, comparing what is in the text of the Results section with Fig. 4d itself, it is unclear what the conclusion really is. The authors need to explain this figure better.

Response:

Thank you for bringing this important issue to our attention and for the positive feedback on the clinical value of our work. We fully agree that a useful feature of the LensAge index is its ability to uncover the risks of eye and other diseases. We have provided analyses of the LensAge index's ability to reveal both ocular aging and systemic diseases (Results section **Page 7 Line 191-192**, **Page 8 Line 193-195** and **Line 215-221**; **Page 9 Line 222**) by comparing the risk of age-related diseases among the participants with a positive LensAge index to that of the participants with a negative LensAge index. Our results demonstrated significantly higher risks of age-related diseases among the participants with a positive LensAge index (**Figure 3 a-b**). In addition, as requested by the Reviewer, we have added the analysis of the differences in risk for individuals with a positive LensAge index above the 75th percentile compared with that for individuals with a LensAge index of "population mean" (between the 25th and 75th percentiles). The updated results have been provided in the revised manuscript (**Page 9 Line 222-227**):

"In addition, individuals with the LensAge index above the 75th percentile had a higher risk of age-related diseases than those with the LensAge index between the 25th and 75th percentiles (diffuse-light mode, adjusted OR=1.61, 95% CI 1.31-1.97, p=5.46e-6; slit-

lamp mode, adjusted OR=1.23, 95% CI 1.00-1.53, p=4.73e-2; Supplementary Table 11).”

More importantly, to investigate the clinical merit of the LensAge index in revealing biological aging, we have performed the ROC curve analysis for the LensAge index versus chronological age in predicting the occurrence of age-related diseases among the participants with a LensAge index less than the 25th percentile or more than the 75th percentile (**Figure c-d** and **Page 9 Line 243-250** and **Page 10 Line 251-254**).

The abovementioned results demonstrate that LensAge was effective for revealing aging status and the individuals with different LensAge indexes have different risks of age-related diseases, indicating that LensAge index has clinical value for uncovering disease risks.

For the smartphone implementation, considering the sample size of analysis dataset, we have conducted comparison of the occurrence of age-related diseases between individuals with positive and negative LensAge index, and analysis of the association of the LensAge index with the risk of occurrence of age-related diseases among individuals with positive LensAge index. The results indicate that LensAge based on smartphone photographs can be an effective indicator of disease risks during aging. The analyzing results have been shown in **Fig. 4d** and in the Results section:

The explanation of the legends of **Fig. 4d**:

“d, The analysis of the LensAge index to reflect age-related diseases based on smartphone photographs among the general population. Analysis (i): comparison of the occurrence of age-related diseases between individuals with positive and negative LensAge index. Analysis (ii): analysis of the association of the LensAge index with the risk of occurrence of age-related diseases among individuals with positive LensAge index.”

The results of smartphone implementation in the Results section (**Page 10 Line 267-275**):

“For the analysis dataset, compared to those with a negative LensAge index based on smartphone photographs, those with a positive LensAge index had a higher risk of age-

related chronic diseases (diabetes, hypertension, coronary heart disease, cancer) (adjusted OR=4.21, 95% CI 1.44-12.36, p=0.009, Fig. 4d). Among these individuals with positive LensAge index, the LensAge index was positively associated with the occurrence of age-related diseases (adjusted OR=1.53, 95% CI 1.09-2.15, p=0.013, Fig. 4d). Thus, LensAge based on smartphone photographs can be an effective indicator of biological age for efficient self-examination of disease risks and health status during aging.”

In the future, a large-scale prospective trial is needed to validate the utility of the smartphone-based determination of LensAge in real-world applications. This statement has been added in the Discussion section (**Page 12 Line 337** and **Page 13 Line 338344**):

“LensAge based on smartphone photographs facilitates routine self-assessment of disease risks and aging/health status. Although our smartphone-based LensAge assessment currently still requires attaching a portable slit lamp for capturing lens photographs, the mHealth mode for the self-monitoring of aging status was shown to be effective. In the future, a large-scale prospective trial is needed to validate the utility of the smartphone-based determination of LensAge in real-world applications, and a more convenient or inexpensive alternative to portable slit lamps must be explored.”

Reviewer #3 (Remarks to the Author):

This study developed a deep learning-based method to predict biological age based on the human lens. The deep learning model is developed on a dataset of 1,990 relatively healthy individuals. Then the model is used to predict the LensAge in the general population including 3,433 participants, where experiments showed that LensAge can reveal the risks of age-related diseases. The method is also validated on 389 smartphone images.

Strengths:

- 1) The research is comprehensive, including the experiment on various age-related diseases and the experiment of different methods.
- 2) The large number and wide types of subjects in this study make the study of high value and credibility.

Response:

We would like to thank the Reviewer for highlighting the strengths of our study.

Major Issues:

- 1) The key rationale for using lens as an indicator of age is not clear. The study expects LensAge to be an objective evaluation method for biological age (line 70). But the interpretation (line 129) of the DL model is not sufficient for it to be an objective method. In line 75, the lens can provide objective age-dependent changes including nucleus enlargement, elasticity reduction, and increased opacity. However, in line 133, the interpretation of the DL model is shown by heatmaps indicating that the DL models focus on the lens. Whether the DL model predicts the biological age based on the key factors of nucleus enlargement, elasticity reduction, and increased opacity still needs to be further discussed. To claim the objectiveness of LensAge, additional experiments are required.

Response:

We initially used Grad-CAM heatmaps to show the regions or structures on which the DL models focused for decision-making. Heatmaps were used to investigate whether the DL model focused on the area of the lens. To further increase the interpretability of the DL model and the objectiveness of LensAge, we have added several analyses, and the updated results have been provided in the revised manuscript:

(1) Analyzing the correlation between lens opacity and LensAge (Page 7 Line 191-192, Page 8 Line 193-194 and Line 195-199)

“Compared with individuals with a negative LensAge index for diffuse-light mode, those with a positive LensAge index had a higher risk of senile cataracts (adjusted OR=1.76, 95% CI 1.45-2.14, $p=1.50e-8$, Fig. 3a).

Among individuals with a positive LensAge index for diffuse-light mode, an increase in the LensAge index was positively associated with the risk of senile cataracts (adjusted OR=1.14, 95% CI 1.10-1.19, $p=5.29e-13$, Fig. 3b). ”

Figure 3. The ability of LensAge to reveal age-related disease risks. a, Comparison of age-related changes between individuals of all ages with positive and negative LensAge indexes for diffuse-light mode or slit-lamp mode. p value from a two-sided test using adjusted logistic regression. b, Association of the LensAge index with age-related

changes in individuals of all ages with a positive LensAge index for diffuse-light mode or slit-lamp mode. *p* value from a two-sided test using adjusted logistic regression.

(2) Analyzing LensAge results for different types of cataracts (Page 5 Line 123-124)

“Moreover, our models demonstrated reasonable performance among different types of cataracts (*Supplementary Table 2*).”

Supplementary Table 2 | LensAge prediction for different types of cataracts

LensAge mode	Cortical cataracts			Nuclear cataracts			Subcapsular cataracts		
	MAE	R ²	p value	MAE	R ²	p value	MAE	R ²	p value
Diffuse-light mode	4.66	0.68	<1.00e-36	4.78	0.64	<1.00e-36	4.51	0.73	4.68e-21
Slit-lamp mode	4.54	0.73	<1.00e-36	5.90	0.58	<1.00e-36	4.38	0.80	5.01e-29

MAE, mean absolute error. *p* value from a two-sided test using linear regression. **p* value<0.05.

(3) Analyzing the importance of different lens structures, including nuclear, cortical, and capsular structures, on the prediction of LensAge results for different types of cataracts (Page 6 Line 141-151)

“To further increase the interpretability of the DL models, we analyzed the influence of masking different lens structures on LensAge prediction for slit-lamp mode. When masking the lens cortex for cortical and noncortical cataracts, the predictive error was higher among cortical cataracts (adjusted odds ratio [OR]=1.23, 95% confidence interval [CI] 1.16-1.31, *p*=6.72e-11, *Supplementary Table 4*), indicating that for this condition, the lens cortex regions were more important for model decision-making than for other types of cataracts. Similarly, the predictive errors were higher among nuclear cataracts and subcapsular cataracts than among other types of cataracts when masking the lens nucleus and lens capsule, respectively (lens nucleus, adjusted OR=1.24, 95% CI 1.16-1.33, *p*=1.55e-10, lens capsule, adjusted OR=1.12, 95% CI 1.02-1.22, *p*=1.60e-2, *Supplementary Table 4*).”

Our findings demonstrate that our model effectively directed its attention towards the specific region associated with the corresponding cataract type for decision-making.

Supplementary Table 4 | Analyses of the importance of different lens structures on LensAge prediction for different types of cataracts

Masking structure	MAE			Adjusted OR (95% CI)	p value
	Cortical cataracts	Nuclear cataracts	Subcapsular cataracts		
Lens cortex	8.47	5.44	4.10	1.23 (1.16-1.31)	6.72e-11*
Lens nucleus	5.95	11.43	4.86	1.24 (1.16-1.33)	1.55e-10*
Lens capsule	5.61	4.25	5.84	1.12 (1.02-1.22)	1.60e-2*

Adjusted logistic regression models were used to compare the influence of masking the cortex on predictive errors between cortical cataracts and other cataracts, the influence of masking the nucleus on predictive errors between nuclear cataracts and other cataracts, and the influence of masking the capsule on predictive errors between subcapsular cataracts and other cataracts. The results are reported with adjusted ORs. MAE, mean absolute error; OR, odds ratio; CI, confidence interval. *p* value from a two-sided test using adjusted logistic regression. **p* value<0.001.

Furthermore, we have added the discussion statement of the key factors for DL model to predict the biological age (**Page 12 Line 313-320**):

“Our interpretation analysis indicates that opacities for different lens structures were important for DL model decision-making, and an increase in the LensAge index was positively associated with senile cataracts, demonstrating that the increased opacities in lens are the key factors for DL model to evaluate the biological age. In addition, in the masking trial to show the importance of different lens structures on LensAge prediction, our model effectively directed its attention towards the specific region associated with the corresponding cataract type, showing the objectiveness of LensAge.”

2) What does LensAge measure? The authors should clarify what is used as ground truth for training the DL model to predict lens ages. If the ground truth is biological age, the authors should specify how the biological age is calculated. If the ground truth is chronological age during the training phase, a concern is whether the DL model truly

learns the biological age or just simply approximates the chronological age. In line 653, the AUC for LensAge and chronological age are close. The authors can consider collecting a precise biological age as described in lines 63-69 for evaluating the precision of the biological age predicted by the DL model. The authors can also consider comparing the LensAge and the chronological age on a group of diseased people to see if chronological age cannot accurately reveal the risks of age-related diseases in this scenario.

Response:

We comprehensively examined the ability of LensAge to accurately represent biological age and predict age-related disease risks, taking multiple steps and not relying solely on chronological age. First, we used lens photographs from a **relatively healthy population** with chronological age labels to train the deep learning models as chronological age predictors to capture the average aging characteristics for biological age reference. Chronological age predictors have shown considerable promise as proxies of the biological age of individuals by learning the population norm in the context of aging [1,2]. In fact, when considering the subsequent question regarding the assessment of biological age using retinal imaging, it is worth noting that two studies mentioned by the Reviewer employed similar methods involving chronological age predictors. Second, the difference between the “LensAge” generated by deep learning models and chronological age was used to predict an individual’s aging level **among the general population** and served as the “LensAge index”. It has been widely hypothesized that the difference between the estimated and actual chronological age, called the ‘ Δ age’ or ‘age gap’, reflects variations in their prior rates of aging [2-4]. These hypotheses have been supported by observations that individuals with positive age gaps, termed age acceleration, are at greater risk of mortality and certain aging diseases, such as heart disease, metabolic disease, and cancers [5-7]. Therefore, in the third step of our study, we analyzed the “LensAge index” to reveal the risk of age-related diseases among the **general population**.

Importantly, to investigate the ability of the LensAge index to reveal biological aging,

we adjusted the regression models for chronological age as a covariate (Method section **Page 19 Line 538**). Subsequent to this adjustment, the LensAge index was still significantly correlated with an increased risk of age-related diseases (**Figure 3 a-b**) and changes in blood glucose (**Supplementary Table 12**). **These findings indicate that LensAge can be an effective proxy of biological age and not just simply approximates the chronological age.**

In our previous manuscript, we conducted ROC curve analysis on the entire population, including patients who aged at an average rate. This implies that chronological age aligned closely with biological age and that the predictive ability for age-related diseases was similar between these two age indexes within this particular group of participants. In response to the Reviewer's request to demonstrate the clinical merit of LensAge, we added ROC curve analysis for the LensAge index versus chronological age in predicting the occurrence of age-related diseases among the participants with a LensAge index less than the 25th percentile or more than the 75th percentile. **These new findings reveal wider gaps in the AUCs, further emphasizing the discriminative power of LensAge in identifying individuals at higher risk of age-related diseases.** The updated results have been provided in the revised manuscript (**Figure c-d and Page 9 Line 243-250 and Page 10 Line 251-254**).

*“We further compared the ability of the LensAge index to predict the occurrence of age-related diseases with that of chronological age using receiver operating characteristic (ROC) curve analysis among the participants with a LensAge index less than the 25th percentile or more than the 75th percentile. The ROC curves graphically illustrate the predictive performance of the LensAge index with an area under the curve (AUC) of 0.621 (95% CI 0.596 to 0.645) for diffuse-light mode (**Fig. 3c**), and 0.600 (95% CI 0.575 to 0.624) for slit-lamp mode (**Fig. 3d**). Compared with those of chronological age, the AUCs of the LensAge index were significantly greater (diffuse-light mode, difference in AUCs=0.098, 95% CI 0.077 to 0.120, Z=8.968, p<0.0001, **Fig. 3c**; slit-lamp mode, difference in AUCs=0.090, 95% CI 0.030 to 0.151, Z=2.924, p=3.50e-3, **Fig. 3d**), demonstrating that LensAge can better reflect the aging process in humans and can be an optimized indicator of age-related disease risks.”*

Figure 3. The ability of LensAge to reveal age-related disease risks. a, Comparison of age-related changes between individuals of all ages with positive and negative LensAge indexes for diffuse-light mode or slit-lamp mode. *p* value from a two-sided test using adjusted logistic regression. **b,** Association of the LensAge index with age-related changes in individuals of all ages with a positive LensAge index for diffuse-light mode or slit-lamp mode. *p* value from a two-sided test using adjusted logistic regression. **c,** Comparison of the AUCs between the LensAge index and chronological age in predicting the occurrence of age-related diseases for diffuse-light mode among the participants with a LensAge index less than the 25th percentile or more than the 75th percentile. LensAge index, AUC=0.621 (95% CI 0.596 to 0.645); chronological age,

AUC=0.523 (95% CI 0.498 to 0.548); difference in AUCs=0.098 (95% CI 0.077 to 0.120), Z=8.968, p<0.0001, n=1,555, two-sided paired DeLong test. d, Comparison of the AUCs between the LensAge index and chronological age in predicting the occurrence of age-related diseases for slit-lamp mode among the participants with a LensAge index less than the 25th percentile or more than the 75th percentile. LensAge index, AUC=0.600 (95% CI 0.575 to 0.624); chronological age, AUC=0.509 (95% CI 0.484 to 0.535); difference in AUCs=0.090 (95% CI 0.030 to 0.151), Z=2.924, p=3.50e-3, n=1,536, two-sided paired DeLong test. CI, confidence interval; ROC, receiver operating characteristic; AUC, area under the curve.

References:

- [1] Galkin F, Mamoshina P, Aliper A, et al. Biohorology and biomarkers of aging: Current state-of-the-art, challenges and opportunities[J]. Ageing Research Reviews, 2020, 60: 101050.
- [2] Rutledge J, Oh H, Wyss-Coray T. Measuring biological age using omics data[J]. Nature Reviews Genetics, 2022: 1-13.
- [3] Hannum G, Guinney J, Zhao L, et al. Genome-wide methylation profiles reveal quantitative views of human aging rates[J]. Molecular Cell, 2013, 49(2): 359-367.
- [4] Horvath S. DNA methylation age of human tissues and cell types[J]. Genome Biology, 2013, 14(10): 1-20.
- [5] Chen B H, Marioni R E, Colicino E, et al. DNA methylation-based measures of biological age: meta-analysis predicting time to death[J]. Aging (Albany NY), 2016, 8(9): 1844.
- [6] Tanaka T, Biancotto A, Moaddel R, et al. Plasma proteomic signature of age in healthy humans[J]. Aging Cell, 2018, 17(5): e12799.
- [7] Levine M E, Lu A T, Quach A, et al. An epigenetic biomarker of aging for lifespan and healthspan[J]. Aging (albany NY), 2018, 10(4): 573.

3) There is some lack of consistency in results in some analysis. In Table 4-7, some p-values>0.1, so the effectiveness of LensAge in these “negative associations” should be

further discussed.

Response:

In the revised manuscript, we have included an additional statement in the Discussion section addressing the negative results (**Page 12 Line 320-329**):

“However, LensAge for slit-lamp mode showed significant correlations only among analyses on certain eye aging diseases and systemic conditions; and analyses of some conditions, such as via ECG and chest X-ray, showed no significant correlation, indicating its weaker ability to reveal aging status compared with that of diffuse-light mode. This phenomenon may occur because of an inconsistent slit-light angle or width, which are important factors in capturing lens photographs and uncovering lens characteristics. In further studies, additional strategies should be implemented to ensure the acquisition of higher quality lens photographs, such as introducing a quality control pipeline to effectively assess and filter out any subpar images.”

4) Novelty of research unclear. There has been some research [1,2] to predict biological age from retinal imaging. The authors should make proper comparisons (quantitative/qualitative) and citations, and state the novelty clearly of LensAge over the previous works.

Reference

[1] Liu, C. et al. Biological Age Estimated from Retinal Imaging: A Novel Biomarker of Aging. In: Medical Image Computing and Computer Assisted Intervention – MICCAI 2019. MICCAI 2019.

[2] Simon N. et al. Retinal photograph-based deep learning predicts biological age, and stratifies morbidity and mortality risk, Age and Ageing, 2022

Response:

We have cited and analyzed different biological age assessments in **Supplementary Table 3**, including the study using retinal images. Importantly, in the two studies stated

by the Reviewer, researchers developed models with chronological age labels as ground truth in similar methods to our current study, **indicating that chronological age predictors can be effective in assessment of biological age.**

Supplementary Table 3 | Performance of different methods for assessing biological age in humans found in the literature

Research	Type of data	Source	Age group (years)	Error (years)	Age-related metric
This study	Lens imaging	Diffuse-light and slit-lamp images	20-96	4.3-4.8	Age-related disease risks
Hannum et al	DNA methylation	Whole blood	19-101	4.9	Age-associated gene expression
Horvath	DNA methylation	Heterogeneous tissues	0-101	3.6	Aging changes in multiple tissues and cell types
Peters et al	Transcriptomics	Peripheral blood mononuclear cells	Mean 28.4-72.2	7.8	Blood pressure, cholesterol levels, fasting glucose, and body mass index
Fleischer et al	Transcriptomics	Dermal fibroblasts	1-94	7.7	Progeria
Putin et al	Blood profiles	Common blood biochemistry and cell count tests	1-100	5.6	Albumin, glucose, alkaline phosphatase, urea, and erythrocytes
Mamoshina et al	Blood profiles	Common blood biochemistry and cell count tests	Interquartile range 32-64	5.9	All-cause mortality risk
Sayed et al	Blood immunome	Peripheral blood mononuclear cells or whole blood	8-96	15.2	Systemic age-related inflammation
Liem et al	Brain imaging	Magnetic resonance imaging (MRI) data	19-82	4.3	Cognitive impairment
Chen et al	3D facial morphologies	3D facial images	17-77	6.1-6.2	Albumin, uric acid, total cholesterol, and low-density lipoprotein cholesterol levels
Zhu et al	Fundus imaging	Retinal fundus and optical coherence tomography images	40-69	3.6	All-cause mortality risk

In response to the Reviewer's request, we have included additional citations in the

Discussion section to highlight the existing methods for assessing biological age based on retinal imaging. Furthermore, we have provided a more explicit statement regarding the novelty of LensAge and its unique contribution to the field. **(Page 11 Line 297-308, Page 12 Line 309-313 and Line 330-337, Page 13 Line 338-344):**

“In addition, some studies have assessed biological age based on ocular retinal photographs [1-3]. A retinal photograph-based model showed better performance [3], but its application is limited by the opaque dioptric media obscuring the fundus, particularly in conditions such as cataracts and vitreous opacity that are common among aged populations. Additionally, capturing retinal images still needs professional equipment and technical requirements for the collecting personnel, which hinders their more widespread and convenient applications.

Biological age estimation based on lens photographs can help to avoid the disadvantages mentioned above and has substantial potential to be applied to self-monitoring and large-scale evaluation. Age-related changes can be objectively observed in the lens, which is transparent under normal conditions and has a connection to the metabolic cycle of the whole body. Additionally, a previous study demonstrated that the lens is essential for the structure and function of the neurological system during aging. The lens serves as a crucial bridge between ocular and systemic aging. In this study, the results indicate that our DL age estimation models learned the lens aging characteristics among populations. LensAge is effective for age evaluation, suggesting that the human lens may be an excellent tissue for monitoring the aging process.

More importantly, lens features can be conveniently and objectively captured by portable devices to identify disease states. Previous assessments of biological age mostly focused on technologies that were not convenient for or available to patients outside the hospital; in other words, they could not be implemented in the community or at home. We applied our smartphone-based method to assess biological age, showing great potential for self-monitoring aging status when needed. Given the rapid development of mobile health (mHealth) and the application of mobile devices, medical

conditions can be monitored and intervened whenever and wherever. LensAge based on smartphone photographs facilitates routine self-assessment of disease risks and aging/health status. Although our smartphone-based LensAge assessment currently still requires attaching a portable slit lamp for capturing lens photographs, the mHealth mode for the self-monitoring of aging status was shown to be effective. In the future, a large-scale prospective trial is needed to validate the utility of the smartphone-based determination of LensAge in real-world applications, and a more convenient or inexpensive alternative to portable slit lamps must be explored.”

References:

- [1] Liu, C. et al. Biological Age Estimated from Retinal Imaging: A Novel Biomarker of Aging. In: Medical Image Computing and Computer Assisted Intervention – MICCAI 2019. MICCAI 2019.
- [2] Simon N. et al. Retinal photograph-based deep learning predicts biological age, and stratifies morbidity and mortality risk, Age and Ageing, 2022
- [3] Zhu, Z. et al. Retinal age gap as a predictive biomarker for mortality risk. British Journal of Ophthalmology (2022).

5) Impact of research unclear. The authors suggest smart phone cameras can measure LensAge. However, whether this is clinically useful is not demonstrated.

Response:

Lens features can be conveniently and objectively captured by portable devices to identify disease states. Previous assessments of biological age typically focused on technologies that were not convenient for or available to patients outside hospitals, limiting their implementation in the community and home. We applied our smartphone-based method to assess biological age, showing its great potential for realizing the self-monitoring of aging status. To investigate the feasibility of the use of smartphones, we initially collected lens photographs using a portable slit lamp (MediWorks portable slit lamp S150, Shanghai) attached to an iPhone/Huawei smartphone (Methods section-

Page 16 Line 423-424). Next, we further assessed whether the LensAge derived from smartphone photographs can be used to evaluate biological age. Nonophthalmologist volunteers used their own smartphones attached to a portable slit lamp to assist patients in capturing lens photographs according to our instructions (**Supplementary Fig. 2**). A total of 389 qualified lens photographs (accounting for 94.4% of all photographs taken) from 102 individuals (mean age [\pm s.d.] of 62.0 [\pm 10.8] years) were obtained using iPhone/Huawei smartphones attached to the portable slit lamp. Among these, 157 images from 50 participants (mean age [\pm s.d.] of 64.6 [\pm 11.4] years) without a medical history of diseases were used as a reference dataset of relatively healthy individuals for accuracy estimation of the DL-age estimation model in slit-lamp mode. Furthermore, the ability of the LensAge index derived from these photographs to reflect the risks of age-related chronic diseases (diabetes, hypertension, coronary heart disease, cancer, cerebral infarction) was investigated with adjusted ORs using logistic regression models (Methods section-**Page 19 Line 515-527**). **These results demonstrate the substantial potential of our smartphone-based LensAge to be conveniently and easily applied in the assessment of biological age and even in the self-monitoring of aging status.**

The baseline characteristics of smartphone datasets are summarized in **Table 1**.

Table 1 | Baseline characteristics of the datasets

	Traditional slit-lamp images		Smartphone images	
	Reference dataset	Analysis dataset	Reference dataset	Analysis dataset
No. of participants	1,990	3,433	50	102
Nationality, n (%)				
Chinese	1,952 (98.1%)	3,370 (98.2%)	50 (100%)	99 (97.1%)
Non-Chinese	38 (1.9%)	63 (1.8%)	0	3 (2.9%)
Age in years (mean\pms.d.)	55.3 \pm 18.0	66.0 \pm 11.5	64.6 \pm 11.4	62.0 \pm 10.8

**Distribution of
chronological age, n
(%)**

>20 and <30	245 (12.3%)	8 (0.2%)	0	0
>30 and <40	231 (11.6%)	55 (1.6%)	3 (6.0%)	4 (3.9%)
>40 and <50	246 (12.4%)	224 (6.5%)	0	6 (5.9%)
>50 and <60	292 (14.7%)	624	15 (30.0%)	
and <70	507 (25.5%)	37 (36.3%)>60 (18.2%) (33.2%)	12 (24.0%)	28 (27.5%)
>70 and <80	334 (16.8%)	976 (28.4%)	18 (36.0%)	23 (22.5%)
>80	134 (6.7%)	407 (11.9%)	2 (4.0%)	4 (3.9%)
Sex, n (%)				
Male	732 (36.8%)	1,482 (43.2%)	20 (40.0%)	45 (44.1%)
Female	1,258 (63.2%)	1,951 (56.8%)	30 (60.0%)	57 (55.9%)
Images, n				
Diffuse-light images	4,542	5,641	N/A	N/A
Slit-lamp images	3,713	5,663	157	389

The results of LensAge for smartphone implementation (**Page 10 Line 256-275**): *“LensAge assessment using smartphones*

We further implemented LensAge to evaluate biological aging using smartphone photographs (Fig. 4a). A total of 389 smartphone images from 102 participants (mean age [\pm s.d.] of 62.0 [\pm 10.8] years, 55.9% females, Table 1) were included in the analysis dataset of smartphone images. Among them, 157 images from 50 participants (mean age [\pm s.d.] of 64.6 [\pm 11.4] years, 60.0% females) without a medical history of diseases were used as a reference dataset of relatively healthy individuals for model accuracy estimation.

The DL model applied to smartphone photographs achieved a strong correlation ($R^2=0.71$ at the individual level, $p=1.59e-14$, Fig. 4b-c) between LensAge and chronological age in the reference dataset, with an MAE of 6.87 years at the image level and 6.80 years at the individual level. For the analysis dataset, compared to those with a negative LensAge index based on smartphone photographs, those with a positive

LensAge index had a higher risk of age-related chronic diseases (diabetes, hypertension, coronary heart disease, cancer) (adjusted OR=4.21, 95% CI 1.44-12.36, p=0.009, Fig. 4d). Among these individuals with a positive LensAge index, the LensAge index was positively associated with the occurrence of age-related diseases (adjusted OR=1.53, 95% CI 1.09-2.15, p=0.013, Fig. 4d). Thus, LensAge based on smartphone photographs can be an effective indicator of biological age for efficient self-examination of disease risks and health status during aging. ”

Figure 4. LensAge estimation using smartphone photographs. a, The application of LensAge on smartphones for self-monitoring aging status. When estimating individuals' aging status using smartphones, lens photographs are collected and sent to the cloud-based AI models. The AI models provide LensAge results, and the results are then sent back to the smartphones. Individuals can take LensAge estimation to self-monitor their aging status when needed. **b,** Scatterplot shows correlation of LensAge at the individual level determined using smartphone photographs with chronological age among relatively healthy participants ($p=1.59e-14$, two-sided linear regression, $n=50$). **c,** Bland-Altman plot shows agreement between LensAge at the individual level using

smartphone photographs and chronological age among relatively healthy individuals. The x-axis represents the mean of LensAge and chronological age, and the y-axis represents the difference between the two measurements (n=50). d, The analysis of the LensAge index to reflect age-related diseases based on smartphone photographs among the general population. Analysis (i): comparison of the occurrence of age-related diseases between individuals with positive and negative LensAge index. Analysis (ii): analysis of the association of the LensAge index with the risk of occurrence of age-related diseases among individuals with positive LensAge index. p value from a two-sided test using adjusted logistic regression. OR, odds ratio; CI, confidence interval.

6) Lack of validation in external dataset. There are lens data from Singapore Epidemiology Studies and the also the US AREDS dataset. It is unclear how applicable is this algorithm outside of Chinese population. The authors may wish to discuss this.

Response:

First, we used the smartphone dataset from other centers to investigate the effectiveness and generalizability of the LensAge models (Methods section **Page 15 Line 417-419**). Our model demonstrated a certain degree of generalizability to other external datasets. Second, we provided a breakdown of the participants' nationalities in the list of baseline characteristics in **Table 1** in the revised manuscript. Although most of the participants enrolled in this study were Chinese, some participants from other nationalities were included in our datasets. The performance of our models in this non-Chinese population shows the potential generalizability of our LensAge models across nationalities (**Supplementary Fig. 1**).

In the current study, our primary goal was to evaluate the feasibility of age-related features in lens to evaluate biological age for a variety of applications and even self-monitoring. In the future, a continuation of this work will expand the study population to other ethnic groups and countries. We have stated this point in the Limitations section (**Page 14 Line 369-373**):

“Second, although a small proportion of non-Chinese people were included in our study,

showing the potential generalizability of our LensAge models across additional ethnicities and nationalities, further larger-scale validation to other ethnic groups and countries is needed as an important continuation of this work.”

Supplementary Figure 1. The LensAge among the non-Chinese population. a, Scatterplot shows the correlation of LensAge at the individual level with chronological age for relatively healthy participants for diffuse-light mode ($p=5.28e-6$, two-sided linear regression, $n=20$). **b,** Scatterplot shows the correlation of LensAge at the individual level with chronological age for relatively healthy participants for slit-lamp mode ($p=1.17e-3$, two-sided linear regression, $n=19$).

REVIEWER COMMENTS

Reviewer #1 (Remarks to the Author):

Our concerns listed in the previous review have been comprehensively addressed and improved on in the revised manuscript.

Reviewer #2 (Remarks to the Author):

In their response to the critique, the authors make the statement that “chronological age predictors have shown considerable promise as proxies of biological age of individuals by learning the population norm in the context of aging.” They also state that “the difference between the “LensAge” generated by deep learning models and chronological age was used to unveil an individual’s aging level among the general population and served as the “LensAge index”. Among the references cited in the response are several that incorporate epigenetic biomarkers. It is abundantly clear that a marker that is a perfect predictor of chronological age is not a good marker of biological age. It is the departure from the population average that constitutes the marker of biological age when the marker is not a perfect predictor of chronological age. In this manuscript, this is the “LensAge index.” The closer to 1.0 the coefficient of determination of the regression of, say, LensAge on chronological age, the less information the biomarker has about biological age over that of chronological age. Any variable correlated with chronological age will of necessity be correlated with any other such variable. Thus, if LensAge or LensAge index are correlated with chronological age, they will be correlated with such variables, and predict age-related disease and degeneration. This is what the authors demonstrate. To escape from this circular logic, some authors demanded that their measures of biological age go further than predicting chronological age by providing the hazard of mortality. The signal importance of doing so was first demonstrated by Kim et al. *GeroScience* (2017) who showed that the so-called DNA methylation clocks of the first and second generations (which included Δ age) were very poor biological age measures in comparison with a frailty index, because they lost all predictive value alongside calendar age. Most subsequent epigenetic clocks have been trained on the hazard of mortality to correct this defect.

The authors, in their response, repeat the same arguments at length twice. Here, and in the manuscript itself, the departure of LensAge from the population mean is at times taken as proof of LensAge Index being a marker of biological age, because such departures from the mean have been assumed to reflect biological aging in the literature. At other places, the properties of LensAge Index are used as an argument in favor of using the departure of a biomarker from the population mean as a measure of biological age. The authors should not have it both ways because they can confuse the uninitiated. The authors have adjusted their regression models for chronological age, and the LensAge Index was still predictive of age-related diseases. With such a large sample, the authors should be able to show data on

its value as a predictor of all-cause mortality. This would go a long way in validating LensAge Index as a metric of biological age, which should be an explicit focus of this manuscript, alongside its clinical utility in the field. LensAge itself should be compared in this regard as well. It is likely that it is not necessary to compute the LensAge Index, after all. The important point that needs more emphasis is that LensAge explains some of the variance in mortality that is not explained by chronological age itself. This discussion should not be relegated to a few passing words in the Methods section.

The authors should pay close attention to their use of LensAge and LensAge Index in the text of the manuscript; they should not be used interchangeably. There are places where LensAge is used as a generic term that encompasses them both. Unless the data are longitudinal, the term pace of aging should not be used.

The arguments regarding the heterogeneity of LensAge Index peaking in middle age and contracting at older ages are very interesting. They are supported by the reference provided in the response by the authors, which deals with the face, and by eye lens (this manuscript). This distinguishes these biomarkers of aging from many others, such as physical function ability and cognitive function, which become progressively heterogeneous with chronological age. This is worth mentioning in the Discussion, and perhaps rationalized.

The authors have added additional analyses for individuals falling within the 25th and above the 75th percentiles to address the critique. This is of minor interest especially since the text does not explain what the purpose of these analyses is.

On page 21, the authors' response summarizes many of the elements that are important in the view of this reviewer. Unfortunately, the authors do not wish to avail themselves of the opportunity to include these points in the manuscript itself. Instead, many analyses that appear unnecessary are presented without a rationale for their inclusion. In other words, the authors did not take the opportunity to interpret their results, for the benefit of readers.

Reviewer #3 (Remarks to the Author):

The authors have made a comprehensive respond to concerns raised by me and other reviewers. The paper has improved

Re: “LensAge index: A deep learning-based biological age for self-monitoring the risks of age-related diseases and mortality” (NCOMMS-23-04543A)

Reviewers’ Comments:

Reviewer #1 (Remarks to the Author):

Our concerns listed in the previous review have been comprehensively addressed and improved on in the revised manuscript.

Response:

We thank the Reviewer for the positive feedback on our revised manuscript.

Reviewer #2 (Remarks to the Author):

In their response to the critique, the authors make the statement that “chronological age predictors have shown considerable promise as proxies of biological age of individuals by learning the population norm in the context of aging.” They also state that “the difference between the “LensAge” generated by deep learning models and chronological age was used to unveil an individual’s aging level among the general population and served as the “LensAge index”. Among the references cited in the response are several that incorporate epigenetic biomarkers. It is abundantly clear that a marker that is a perfect predictor of chronological age is not a good marker of biological age. It is the departure from the population average that constitutes the marker of biological age when the marker is not a perfect predictor of chronological age. In this manuscript, this is the “LensAge index.” The closer to 1.0 the coefficient of determination of the regression of, say, LensAge on chronological age, the less information the biomarker has about biological age over that of chronological age. Any variable correlated with chronological age will of necessity be correlated with any other such variable. Thus, if LensAge or LensAge index are correlated with chronological

age, they will be correlated with such variables, and predict age-related disease and degeneration. This is what the authors demonstrate. To escape from this circular logic, some authors demanded that their measures of biological age go further than predicting chronological age by providing the hazard of mortality. The signal importance of doing so was first demonstrated by Kim et al. *GeroScience* (2017) who showed that the so-called DNA methylation clocks of the first and second generations (which included Δ age) were very poor biological age measures in comparison with a frailty index, because they lost all predictive value alongside calendar age. Most subsequent epigenetic clocks have been trained on the hazard of mortality to correct this defect.

The authors, in their response, repeat the same arguments at length twice. Here, and in the manuscript itself, the departure of LensAge from the population mean is at times taken as proof of LensAge Index being a marker of biological age, because such departures from the mean have been assumed to reflect biological aging in the literature. At other places, the properties of LensAge Index are used as an argument in favor of using the departure of a biomarker from the population mean as a measure of biological age. The authors should not have it both ways because they can confuse the uninitiated. The authors have adjusted their regression models for chronological age, and the LensAge Index was still predictive of age-related diseases. With such a large sample, the authors should be able to show data on its value as a predictor of all-cause mortality. This would go a long way in validating LensAge Index as a metric of biological age, which should be an explicit focus of this manuscript, alongside its clinical utility in the field. LensAge itself should be compared in this regard as well. It is likely that it is not necessary to compute the LensAge Index, after all. The important point that needs more emphasis is that LensAge explains some of the variance in mortality that is not explained by chronological age itself. This discussion should not be relegated to a few passing words in the Methods section.

Response:

Thank you for bringing these important concerns to our attention. Initially, we employed a relatively healthy population to develop our deep learning (DL) models. In

this stage, models were expected to learn the average aging characteristics of the relatively healthy population as reference; therefore, we conducted correlation analyses to evaluate the model fitness. While in the subsequent stage, we expanded our analyses to encompass a general population consisting of both healthy and unhealthy participants. We aimed to measure individual's aging acceleration or deceleration relative to the population average learned on the relative healthy cohort. Therefore, we calculated the deviations. Indeed, different groups of participants were included in the stages of model development and model testing, which may help address the circular logic concern raised by the Reviewer.

Aging is a multifaceted process, encompassing various aspects such as the decline of bodily functions, the occurrence and progression of age-related diseases, and mortality [1,2]. In our previous manuscript, our primary focus was on assessing risks of age-related diseases. This choice was motivated by the vital potential use of our models for self-monitoring health status across different age groups. In this context, the practical significance of evaluating the risks of disease occurrence may outweigh mortality prediction, particularly among younger individuals with extremely low mortality rates. Hence, we conducted investigations to examine our models' ability to reflect biological age and the risks of age-related diseases, rather than solely reflecting the passage of time. Importantly, after adjusting for chronological age and other covariates in regression models, the LensAge index still demonstrated significant correlations with increased risks of age-related diseases and indicators. In addition, we conducted the ROC curve analyses to show the superior ability of the LensAge index compared to chronological age for assessing the risks of age-related diseases.

Furthermore, in response to the Reviewer's request and to comprehensively assess our models, we followed up the individuals in the analysis dataset to the latest month for survival outcome. After a median follow-up of 30.2 months (IQR 28.0-32.0 months), a total of 66 (2.2%) participants died from all causes. We employed Cox proportional hazards regression models to evaluate the predictive performance of the LensAge index for all-cause mortality, while adjusting for chronological age, sex, and other covariates. In addition, we compared all-cause mortality of participants by different quartiles of the

LensAge index with the lowest quartile as reference. The results show that for the diffuse-light mode, each 1-year increase in the LensAge index was associated with an 8% relative increase in the risk of all-cause mortality (adjusted hazard rate [HR]=1.08, 95% CI 1.01-1.15, $p=2.43e-2$; **Supplementary Table 13**). Compared to participants in the lowest quartile of the LensAge index, those in the second quartile had comparable mortality risk (adjusted HR=1.09, 95% CI 0.46-2.60, $p=8.38e-1$; **Fig. 4a**). Notably, participants in the third and highest quartile had significantly increased mortality risks compared to those in the lowest quartile (adjusted HR=2.53, 95% CI 1.20-5.30, $p=1.44e-2$; adjusted HR=2.94, 95% CI 1.38-6.27, $p=5.10e-3$; respectively; **Fig. 4a**). Similar results were observed for the survival analyses conducted using the slit-lamp mode. Our results indicate that the LensAge index serves as a noteworthy predictor of all-cause mortality, elucidating a portion of the mortality variation that remains unexplained by chronological age itself. This highlights its significant clinical value as an indicator of biological age in identifying individuals at high risk of mortality, enabling early interventions.

The updated methods and results have been provided in the revised manuscript:

Results-Page 10 Line 265-280 and Page 11 Line 281-282

“The predictive performance of the LensAge index for all-cause mortality

*To comprehensively assess the LensAge index’s ability to indicate biological age, we evaluated the predictive performance of the LensAge index for all-cause mortality using Cox proportional hazards regression models adjusted for chronological age, sex, race, region, occupation, smoking, and alcohol intake status. The results show that each 1-year increase in the LensAge index for the diffuse-light mode was associated with an 8% relative increase in the risk of all-cause mortality (adjusted hazard rate [HR]=1.08, 95% CI 1.01-1.15, $p=2.43e-2$; **Supplementary Table 13**). Comparatively, individuals in the second quartile of the LensAge index had a similar mortality risk to those in the lowest quartile (adjusted HR=1.09, 95% CI 0.46-2.60, $p=8.38e-1$; **Fig. 4a**). Notably, participants in the third and fourth quartiles of the LensAge index had significantly increased all-cause mortality risks compared to those in the lowest quartile (adjusted HR=2.53, 95% CI 1.20-5.30, $p=1.44e-2$; adjusted HR=2.94, 95% CI 1.38-6.27,*

$p=5.10e-3$; respectively; **Fig. 4a**). Similar results were observed for the survival analyses conducted using the slit-lamp mode (**Fig. 4b** and **Supplementary Table 13**). These findings demonstrate that the LensAge index is a significant predictor of survival, and the acceleration or deceleration of aging detected by LensAge measurements aligns with individual aging status.”

Methods-Page 20 Line 565-570 and Page 21 Line 571-582

“Evaluation of the predictive performance of the LensAge index for all-cause mortality

*The participants in the analysis dataset were followed up from the time when the lens photographs were taken. To gather information on all-cause mortality status and date of death, questionnaires were administered by the investigators to the relatives of the participants. The duration of follow-up for each participant was calculated as the time elapsed between their baseline and the date of death or the completion of the follow-up period (July, 2023), whichever came first. The percentage of individuals in the analysis dataset lost to follow-up was 13.1%. After a median follow-up of 30.2 months (IQR 28.0-32.0 months), a total of 66 (2.2%) individuals died from all causes. The baseline information of the follow-up and loss to follow-up individuals was comparable (**Supplementary Table 15**). To assess the predictive performance of the LensAge index for all-cause mortality risk, Cox proportional hazards regression models adjusted for chronological age, sex, race, region, occupation, smoking, and alcohol intake status were utilized. These models estimated the impact of a 1-year increase in the LensAge index on the risk of all-cause mortality. Additionally, we compared all-cause mortality of participants in the different quartiles of the LensAge index with those in the lowest quartile for reference.”*

In addition, we have added the statement of the predictive ability of our models for the mortality risk in the Discussion (**Page 14 Line 396** and **Page 15 Line 397-417**):

“Importantly, the process of aging is multifaceted, involving various aspects such as the decline of bodily functions, the occurrence and progression of age-related diseases, and mortality. In our study, we extended the application of our models to evaluate age-related status and the risk of all-cause mortality in the general population. By adjusting

for chronological age as a covariate, we found that the LensAge index was significantly correlated with an increased risk of age-related diseases in human eyes. Additionally, we also provided evidences for potential associations between lens aging and systemic aging. Notably, the LensAge index showed significantly correlated with age-related chronic diseases and BG levels, shedding light on the metabolic conditions of individuals. Furthermore, previous studies have applied markers of biological age to predict mortality or estimate time until death, highlighting the potential utility of such markers. Similarly, through a comprehensive assessment of the predictive performance of the LensAge index for mortality risk, our results indicate that the LensAge index serves as a noteworthy predictor of all-cause mortality in humans, elucidating a portion of the mortality variation that remains unexplained by chronological age itself. This highlights its significant clinical value as an indicator of biological age identifying individuals at high risk of mortality, enabling early interventions. These findings suggest a considerable synchronization between the aging processes in the lens and systemic metabolism, extending to the mortality risk. Therefore, the application of our lens aging assessment methods hold promise for self-monitoring health status and targeted interventions aimed at addressing aging-related concerns.”

Supplementary Table 13 | Analysis of the predictive performance of the LensAge index for all-cause mortality

LensAge index	A d j u s t e d 9 5 % C I			p value
	HR	Lower	Upper	
Diffuse-light mode	1.08	1.01	1.15	2.43e-2*
Slit-lamp mode	1.07	1.01	1.12	1.21e-2*

HR=hazard rate; CI=confidence interval. *p* value from a two-sided test using adjusted Cox proportional hazards regression. **p* value<0.05.

Supplementary Table 15 | Comparison of baseline characteristics of follow-up and lost to follow-up

	Follow-up	Loss to follow-up	p value
No. of participants	2,982	451	-
Chronological age in years (mean±s.d.)	65.9±11.5	66.7±11.3	0.19 ^a
Sex, n (%)			
Male	1,287	195	0.98 ^b
Female	1,695	256	
LensAge index in years (mean±s.d.)			
Diffuse-light mode	2.0±5.9	2.3±5.9	0.32 ^a
Slit-lamp mode	2.8±7.1	2.9±6.8	0.79 ^a

^a*p* value from a two-sided test using Student's *t*-test; ^b*p* value from a two-sided test using χ^2 test.

Figure 4. Survival curves for all-cause mortality risk by LensAge index quartiles. Mortality risks are shown over time for participants in different LensAge index quartiles for diffuse-light mode (a) and slit-lamp mode (b). Lower quartiles correspond to individuals who had smaller LensAge index, whereas higher quartiles correspond to those with greater LensAge index. Adjusted HR values were calculated using Cox proportional hazards regression models, adjusted for chronological age, sex, race, region, occupation, smoking, and alcohol intake status. Comparatively, individuals in the second quartile of the LensAge index had a similar mortality risk to those in the lowest quartile ($p>0.05$). Individuals in the third and fourth quartiles of the LensAge index had significantly relative increased all-cause mortality risks compared to those in the lowest quartile ($p<0.05$). HR, hazard ratio.

References:

[1] Rutledge J, Oh H, Wyss-Coray T. Measuring biological age using omics data[J]. Nature Reviews Genetics, 2022: 1-13.

[2] Tian Y E, Cropley V, Maier A B, et al. Heterogeneous aging across multiple organ systems and prediction of chronic disease and mortality[J]. Nature Medicine, 2023, 29(5): 1221-1231.

The authors should pay close attention to their use of LensAge and LensAge Index in the text of the manuscript; they should not be used interchangeably. There are places where LensAge is used as a generic term that encompasses them both. Unless the data are longitudinal, the term pace of aging should not be used.

Response:

The LensAge index holds practical significance in self-monitoring health status as it reflects an individual's accelerated or decelerated aging relative to peers of the same chronological age. In response to the Reviewer's request, we have given more attention to the usage of LensAge index compared to LensAge and revised the corresponding words throughout the revised manuscript, ensuring that their usage is not interchangeable. Furthermore, we have removed the term "pace of aging" to enhance clarity and precision (**Page 7 Line 165-167**).

The arguments regarding the heterogeneity of LensAge Index peaking in middle age and contracting at older ages are very interesting. They are supported by the reference provided in the response by the authors, which deals with the face, and by eye lens (this manuscript). This distinguishes these biomarkers of aging from many others, such as physical function ability and cognitive function, which become progressively heterogeneous with chronological age. This is worth mentioning in the Discussion, and perhaps rationalized.

Response:

We thank the Reviewer for raising this issue. We have added the statement of this point in the Discussion (**Page 14 Line 378-395**):

“Given that age-related changes occur throughout the entire lifespan and manifests at different rates across different age groups, we investigated a more diverse population spanning ages 20 to 96 years to provide a more comprehensive understanding. Our findings specifically highlight the significant disparity of the LensAge index among individuals younger than 60 years old. This observation aligns with previous studies investigating the aging process, which have consistently reported smaller variations and more stable states of specific biological age makers among individuals older than 60 years [1,2]. Consequently, it suggests that interventions targeting aging should primarily be initiated during the stage characterized by substantial heterogeneity in the context of aging. Nevertheless, it is important to note that the clinical merit of our models is not diminished within the older age groups. The LensAge index provides a measure of an individual’s aging level relative to their peers of the same chronological age. In the population aged more than 60 years, a positive LensAge index showed a significant correlation with an increased risk of age-related diseases (Supplementary Table 6), demonstrating the effectiveness of our models in accurately identifying disease risks within this age group. Therefore, our method provides an effective indicator for revealing the biological age of populations spanning different age groups.”

References:

1. Xia, X. et al. Three-dimensional facial-image analysis to predict heterogeneity of the human ageing rate and the impact of lifestyle. *Nature metabolism* 2, 946-957 (2020).
2. Zhu, Z. et al. Retinal age gap as a predictive biomarker for mortality risk. *British Journal of Ophthalmology* (2022).

The authors have added additional analyses for individuals falling within the 25th and above the 75th percentiles to address the critique. This is of minor interest especially since the text does not explain what the purpose of these analyses is.

Response:

We conducted ROC curve analyses to demonstrate the superior ability of the LensAge index compared to chronological age in evaluating the risks of age-related diseases. In the initial version of the manuscript, our focus was on the entire population. While the LensAge index exhibited relatively better capabilities in assessing age-related diseases compared to chronological age, the difference in their AUC values was relatively small. To further evaluate the clinical value of our models, we extended our analyses to include participants in the lowest and highest quartiles of the LensAge index, suggesting significant deceleration and acceleration of lens aging, respectively. The additional analyses revealed a larger gap in AUC values. This further confirms the LensAge index's effectiveness in assessing the risk of age-related diseases and its utility as a robust biological age indicator, particularly evident in individuals with notable acceleration or deceleration of lens aging. Importantly, our updated analyses of mortality risk consistently demonstrate a significant increase in all-cause mortality risk among individuals in the highest quartile of the LensAge index compared to those in the lowest quartile.

To clarify the purpose of the ROC curve analysis among the participants with a LensAge index less than the 25th percentile or more than the 75th percentile, we have added the clearer statement in the revised manuscript (**Page 9 Line 251 and Page 10 Line 252-255**):

“We further compared the ability of the LensAge index to predict the occurrence of age-related diseases with that of chronological age using receiver operating characteristic (ROC) curve analyses, particular among individuals with a significantly greater LensAge index. We performed the ROC analyses among the participants with a LensAge index in the lowest quartile or the highest quartile.”

On page 21, the authors' response summarizes many of the elements that are important in the view of this reviewer. Unfortunately, the authors do not wish to avail themselves of the opportunity to include these points in the manuscript itself. Instead, many

analyses that appear unnecessary are presented without a rationale for their inclusion. In other words, the authors did not take the opportunity to interpret their results, for the benefit of readers.

Response:

We thank the Reviewer for bringing up this concern. We apologize for any confusion caused by our previous response. In order to show the clinical merit of our method, we comprehensively analyzed the ability of the LensAge index to reflect the risk of both diseases and mortality (updated analyses in the revised manuscript). Importantly, all the regression models used in our analyses were adjusted for chronological age as a covariate. This adjustment aimed to investigate whether the LensAge index can serve as an effective indicator of biological age, going beyond its role as a mere proxy for chronological age. Our results reveal a significantly positive association between an increase in the LensAge index and the risks of both diseases and all-cause mortality. This association effectively explains a substantial portion of the variation in mortality and disease risks that remains unexplained by chronological age alone, providing valuable insights into biological age.

Furthermore, we have added the clearer statements for better interpretation of the purpose and conclusions of our analyses in the revised manuscript:

For the interpretation of the adjustment for chronological age in regression models (**Page 7 Line 190-193** and **Page 8 Line 194**):

“In order to investigate whether the LensAge index can be an effective marker of biological age, we assessed the its ability to reveal the risks of ocular age-related conditions in the analysis dataset using logistic models adjusted for demographic (chronological age, sex, race, region, and occupation) and lifestyle (smoking and alcohol intake status) covariates.”

For the interpretation of analyzing the differences in risk of age-related diseases for individuals with a positive LensAge index compared with that for individuals with a negative LensAge index (**Page 8 Line 212-214** and **Page 10 Line 262-263**):

“These findings indicate that the LensAge index can be an effective measure for

biological age to reflect risks of ocular age-related diseases and advanced aging in the human eyes.”

“the LensAge index can better reflect the aging process in humans and can be an optimized indicator of age-related disease risks in whole body.”

For the interpretation of the analysis of the differences in risk of age-related diseases between individuals with a LensAge index above the 75th percentile and individuals with a LensAge index at the moderate level (between the 25th and 75th percentiles) (**Page 9 Line 230-233**):

“In addition, to further evaluate the impact of significantly higher LensAge index on biological aging, we compared the differences in the risks of age-related diseases between individuals with a LensAge index in the highest quartile and at the moderate level (in the second or third quartiles).”

Reviewer #3 (Remarks to the Author):

The authors have made a comprehensive respond to concerns raised by me and other reviewers. The paper has improved

We thank the Reviewer for the positive feedback on our revised manuscript.

REVIEWERS' COMMENTS

Reviewer #2 (Remarks to the Author):

The authors have answered most concerns satisfactorily. This is a much improved manuscript.

Re: “LensAge index: A deep learning-based biological age for self-monitoring the risks of age-related diseases and mortality” (NCOMMS-23-04543B)

Reviewers’ Comments:

Reviewer #2 (Remarks to the Author):

The authors have answered most concerns satisfactorily. This is a much improved manuscript.

Response:

We thank the Reviewer for the positive feedback on our revised manuscript.